# Birch leaves and branches as a source of ice-nucleating macromolecules

Laura Felgitsch[1], Philipp Baloh[1], Julia Burkart[1], Maximilian Mayr[1], Mohammad E. Momken[1], Teresa M. Seifried[1], Philipp Winkler[1], David G. Schmale III[2], Hinrich Grothe[1]

[1]Institute of Materials Chemistry, TU Wien, Vienna, 1060, Austria
[2]Department of Plant Pathology, Physiology, and Weed Science, Virginia Tech, 24061-0390 Blacksburg, Virginia, USA

*Correspondence to*: Hinrich Grothe (grothe@tuwien.ac.at)

**Abstract.** Birch pollen are known to release ice-nucleating macromolecules (INM), but little is known about the production and release of INM from other parts of the tree. We examined the ice nucleation activity of samples from ten different birch trees (*Betula spp.*). Samples were taken from nine birch trees in Tyrol, Austria, and from one tree in a small urban park in Vienna, Austria. Filtered aqueous extracts of 30 samples of leaves, primary wood (new branch wood, green in colour, photosynthetically active), and secondary wood (older wood of a branch, brown in colour, with no photosynthetic activity) were analysed in terms of ice nucleation activity using VODCA (Vienna Optical Droplet Crystallization Analyser), a cryo microscope for emulsion samples. All samples contained ice nuclei in the submicron size range. Concentrations of ice nuclei ranged from $6.7*10^4$ to $6.1*10^9$ per mg sample. Mean freezing temperatures varied between -15.6 °C and -31.3 °C; the range of temperatures where washes of birch pollen and dilutions thereof typically freeze. The freezing behaviour of three concentrations of birch pollen washing water (initial wash, 1:100, and 1:10,000), were significantly associated with more than a quarter of our samples, including some of the samples with highest and lowest activity, indicating a relationship between the INM of wood, leaves and pollen. Extracts derived from secondary wood showed the highest concentrations of INM and the highest freezing temperatures. Extracts from the leaves exhibited the highest variation in INM and freezing temperatures. Infrared spectra of the extracts and tested birch samples show qualitative similarity, suggesting the chemical components may be broadly similar.

## 1 Introduction

Pure water can typically be supercooled to temperatures below the melting point of ice (0 °C at atmospheric pressure) without freezing (Cantrell and Heymsfield, 2005; Hegg and Baker, 2009; Murray et al., 2010). In order to freeze, water molecules have to be arranged in an ice like pattern and overcome a critical cluster size (Turnball and Fisher, 1949; Cantrell and Heymsfield, 2005). This freezing mechanism, if happening as a stochastic process from the pure liquid, and in the absence of catalysing substances, is called homogeneous ice nucleation (Cantrell and Heymsfield, 2005). In micrometre-sized droplets this phase change takes place at temperatures below -35 °C (Pruppbacher and Klett, 1997). However, freezing can also be triggered at higher sub-zero temperatures by foreign substances (Dorsey, 1948) called ice nucleating particles (INP, Vali et al., 2015), which is referred to as heterogeneous freezing. In the atmosphere, INP can contribute to cloud glaciation and precipitation (Lohmann, 2002). Ice clouds impact the radiation balance of the Earth and therefore our climate (Mishchenko et al., 1996; Baker, 1997; Lohmann, 2002; Intergovernmental Panel on Climate Change, 2007). Representatives of many different substance classes of aerosols have been found to act as INP (Hoose and Möhler, 2012; Murray et al., 2012). Despite this ubiquitous distribution throughout different aerosol species, ice nucleation active material only represents a small part of total atmospheric aerosol (Rogers et al., 1998). Typical total aerosol concentrations range between $10^2$ and $10^3$ per $cm^3$ for free troposphere and marine boundary layer concentrations, and between $10^3$ and $10^5$ per $cm^3$ for continental boundary layer concentrations (Spracklen et al., 2010). INP concentrations are much lower and range between $10^{-1}$ and $10^{-4}$ per $cm^3$ (Rogers et al., 1998; DeMott et al., 2010).

There are significant gaps in the understanding of heterogeneous ice nucleation and the contributions of different sources of INP. The role of biological substances in this process is understudied (Möhler et al., 2007; Murray et al., 2012). Field studies have demonstrated that the biosphere acts as an important source for primary aerosol particles (Jaenicke, 2005). Jia et al. (2010) analysed carbon sources of PM2.5 particles (particulate matter with an aerodynamic diameter of 2.5 μm or smaller) collected at an urban and a rural site in Texas, and attributed 5-13 % of the particle mass to primary biological sources and 4-9 % to secondary organic aerosols. Biological residues can be adsorbed on dust particles (O'Sullivan et al., 2016). Even small amounts of adsorbed biological matter can increase nucleation temperatures of less active ice nuclei (Conen et al., 2011). Several studies point to the importance of biological material in cloud processes. Precipitation can contain large amounts of INP. Petters and Wright (2015) combined data from a large number of measurements and found a high variability in concentration in the range between -5 and -12 °C, which is assumed to be biological, with a maximum of approx. 500 000 INP per L water. Christner et al. (2008) analysed snow and rain samples from the United States (Montana and Louisiana), the Alps and the Pyrenees, Antarctica (Ross Island) and Canada (Yukon), where they found rather low INP concentrations, but biological INP to represent the majority of the contained INP. Pratt et al. (2009) examined ice crystal residues collected from ice clouds 8 km over Wyoming, US, and about a third of the collected material was of biological origin. Moreover, 60 % of the highly abundant mineral dusts were internally mixed with biological or humic substances. Kamphus et al. (2010) analysed ice crystal residues from mixed phase clouds at the Jungfraujoch station in the Swiss Alps, and found that 2-3 % of the material at 3500 m could be classified as biological. Conen et al. (2016) found indications that leaf litter, which naturally hosts a vast variety of microorganisms, enriches Arctic air with ice nucleating particles. Huffman et al. (2013) collected aerosols above woodlands in Colorado. They observed a burst in biological INP concentrations in the atmosphere that appeared to be linked to rain events. Since biological INP are capable of influencing cloud glaciation and precipitation (Sands et al., 1982; Morris et al., 2014), rain-induced bursts might be important contributors to atmospheric and hydrological processes.

Biological material from plants could be an abundant source of INP. The controlled freezing of water within a plant is an important mechanism for plants to cope with cold climatic conditions. The freezing of water is challenging for living organisms, since it often leads to lethal injuries during the process (Storey and Storey, 2004). Some plants that are exposed to cold stress have developed unique strategies to ensure their survival (Zachariassen and Kristiansen, 2000). Intracellular freezing can lead to a disruption of the cell and typically has lethal consequences for the cell and subsequently for the plants (Mazur, 1969; Burke et al., 1976; Pearce, 2001). Many plants grow in climatic zones where temperatures regularly fall low enough to make a complete avoidance of freezing impossible. To avoid cell damage under such conditions, those plants typically trigger the freezing process in their extracellular spaces (Burke et al., 1976), a process that can be achieved by releasing INP in the plant's tissue. This freezing process leads to a dehydration of the cell, due to the attraction of intracellular water by extracellular ice (Mazur, 1969). Dehydration induces several changes inside of cells such as changes in pH-value, salt concentration, and protein denaturation. Therefore, frost hardiness is often defined by the degree of dehydration a plant can survive (Burke et al., 1976). During cell dehydration, a rapid increase in concentration of ions and small molecules inside the cell takes place, leading to freezing point depression and thus hinders intracellular ice formation (Burke et al., 1976). If temperatures fall too low, the high intracellular salt concentration often promotes glass formation (Hirsh et al., 1985). Frost hardy plants are able to survive rapid cooling to liquid nitrogen temperatures, if they are pre-frozen at -15 to -30 °C depending on the plant and time of the year (Sakai, 1973). These results show that controlled freezing can be an important mechanism for plants to cope with cold climatic conditions. Though even controlled freezing comes with a risk for plants (e.g. cavitation due to bubble formation (Sperry and Sullivan, 1992)), many plants have been found to be ice nucleation active. Such plants are e.g. blueberry (Kishimoto et al., 2014), sea buckthorn (Jann et al., 1997; Lundheim and Wahlberg, 1998), and winter rye (Brush et al., 1994). These processes and findings indicate that plants are a viable source of INP, a topic that requires further study.

Spectroscopic methods are a key instrument in characterizing complex biological systems. One of the methods typically applied on biological materials is infrared spectroscopy (Baker et al., 2015), which allows characterization and discrimination of plants (Kim et al., 2004; Gorgulu et al., 2007; Anilkumar et al., 2012; Carballo-Meilan et al., 2014). Further, infrared spectroscopy has already shown to respond well on the biochemical features of pollen of different species, which allows differentiation of such (Gottardini et al., 2007; Pummer et al., 2013; Zimmermann and Kohler, 2014; Bağcioğlu et al., 2015). It can even be used to gain information on the environmental conditions (Zimmermann and Kohler, 2014).

In our study we look for INP in different parts of birch trees. Birch pollen are already known to exhibit ice nucleation activity (INA) (Diehl et al., 2001), and recent research suggests that pollen grains play a role in local INP concentrations during pollen peak periods (Kohn, 2016). They easily release their ice nucleation active compounds which are in the macromolecular size range (Pummer et al., 2012). However, little is known about the production and release of these ice-nucleating macromolecules (INM) from other parts of the tree. We hypothesized that the materials throughout birch trees are ice nucleation active and that the active compound(s) in these birch materials from different parts of the tree are similar to those in birch pollen. The specific objectives of this study were to (1) investigate the INA of the different birch tree samples, especially in regard to similarities to the INM, which have already been found in birch pollen (Pummer et al., 2012, 2015), (2) determine the distribution of INM throughout leaves and branches of birch trees, and (3) compare spectroscopic and ice nucleation results of different birch trees to establish the variability in chemical nature and INA of the different trees.

## 2 Materials and methods

### 2.1 Samples

Samples were collected from nine birches in Tyrol, Austria (named TB for Tyrolian Birch and numbered A to I) and one birch located in an urban park in Vienna, Austria in the spring and summer of 2016. Detailed descriptions of all investigated birches can be found in

Table 1. Larger branches were removed from the lower 3 m of the canopy, and were divided into three sample groups including leaves, ~5 cm sections of primary wood (green, photosynthetically active), and ~5 cm sections of secondary wood (brown, no photosynthetic activity). Representative material was combined for each tree, resulting in thirty bulk samples (1 bulk sample of each category, per tree) for downstream analyses. All tools used were surface disinfected with 90 % ethanol prior to branch removal. The samples were stored in a cooler for transport back to the laboratory, and were frozen within a few hours of collection at -20 °C. The Tyrolian samples were collected along an altitudinal gradient (from altitudes between 799 m to 1,925 m). The locations of the Tyrolian birches are shown in Figure 1. Birch pollen used for FTIR spectroscopy were *Betula pendula* pollen from AllergonAB (Thermo Fisher).

### 2.2 Sample preparation

Samples were processed using the following milling procedure. Prior to milling, visible contaminations on the outside of the samples (e.g., lichens) were removed. A swing mill (Retsch MM400) was used (with a frequency of 25 s$^{-1}$) to mill each of the samples. We used approx. 20 cm increments per wood sample (cut into pieces of about 0.5 cm) and 2-3 leaves per leaf sample, which were milled and bulked together. In all cases the wood and leaf samples stemmed from a single branch per tree. Each sample was cooled with liquid nitrogen between two milling steps. We achieved this by immersing the milling container containing the sample and the ball (stainless steel) in liquid nitrogen. After equilibrium was reached, we remounted the container on the mill and conducted the next milling step. We milled each sample four times for 30 s. After the milling process the products were dried in vacuum over silica gel until the weight was constant. All samples were dried for at least twelve hours. Weight consistency was determined by two weighing steps separated by at least two hours of drying. Total dry mass of the sample bulks varied between approx. 100 and 600 mg. Part of the dried bulk was immersed in ultrapure water (produced with Millipore® SAS SIMSV0001) (1 ml per 50 mg of powder). Over a time of six hours the mixture was shaken

two to four times. Afterwards it was centrifuged (3500 rpm/ 1123 g for 5 min) and the supernatant was pressed through a 0.2 µm syringe filter (VWR, cellulose acetate membrane, sterile), removing all bigger particles, as well as possible impurities e.g. intact bacterial cells.

Birch pollen washing water was prepared using 50 mg pollen and adding 1 ml of ultrapure water. The suspension was treated the same way as the wood and leaf suspensions, except for centrifuging, which was done for 10 min. Since we filtered our samples, all data presented refers to INM concentrations in the submicron size range (per mg sample mass, extractable aqueously with a 50 mg/mL sample load within six hours).

## 2.3 VODCA (Vienna Optical Droplet Crystallisation Analyser)

The Vienna Optical Droplet Crystallisation Analyser (VODCA) was used to determine INA as described by Pummer et al (2012). To monitor freezing of separated droplets, emulsions were created consisting of an aqueous phase in paraffin oil containing lanolin as emulsifier. As aqueous part of the emulsion ultrapure water was used for blank measurements and sample extracts were used for sample measurements. The emulsions were prepared on thin glass slides via mixing by hand with a pipette tip with oil in small excess, leading to aqueous droplets in an inert phase (Hauptmann et al., 2016). One glass slide was then placed on a Peltier element (Quick-cool QC-31-1.4-3.7M) with a thermocouple on its surface (next to the sample spot). The Peltier element was mounted on a copper cooling block cooled by an ice water cycle. The element and the cooling block were situated in an air tight cell, which was closed during measurements. To prevent humidity from interfering with measurements, the cell was flushed with dry nitrogen gas whenever the sample was changed. To observe the freezing events we used an incident light microscope (Olympus BX51M) with an attached camera (Hengtech MDC320) linked to a computer.

Once the sample had been placed on the Peltier element and the cell was closed, the cooling process was started. All here presented data was obtained with a cooling rate of 10 °C/min. To evaluate freezing, photos were taken during the whole process. The first one was always taken of the unfrozen sample as a blank. For each photo the respective sample temperature, $T_{photo}$, was recorded. Comparison of different photos made it possible to evaluate the number of frozen droplets and therefore the frozen droplet fraction at a certain temperature. Cooling continued until all droplets were frozen. Only droplets in the size range between 15 and 40 µm (droplet volume: 1.8 – 34 pL) were included in our evaluation.

## 2.4 Data analysis

Results of the freezing experiments are presented as cumulative nucleus concentration (see below) and as mean freezing temperature (MFT). The MFT is the weighted average freezing temperature of all analysed droplets of a single aqueous sample extract, determined by the following the equation:

$$MFT = \frac{\sum T_i * n_i}{n_{-35°C}} \tag{1}$$

with $T_i$ being a recorded temperature, $n_i$ being the number of droplets freezing at this temperature, and $n_{-35°C}$ being the number of droplets frozen at temperatures of -35 °C and higher. The formula only accounts for temperatures of -35 °C and higher and consequently only for droplets frozen at these temperatures. This is done to minimize the risk of including homogeneous freezing events in our presented data.

The cumulative nucleus concentration $K(T_{photo})$ was used as an indicator for the number of INM at temperatures above $T_{photo}$ contained in the sample. To determine IN concentrations, the number of frozen droplets $n_{frozen}$ for a given temperature $T_{photo}$ were counted. The droplet volume included in the evaluation was calculated for a droplet with a diameter of 25 µm (median droplet diameter). To prevent an underestimation of the concentration of INM freezing at lower temperatures (Govindarajan

and Lindow, 1988), samples showing no homogeneous freezing in the first measurement were diluted and re-measured. The measurements of diluted samples were only used for the determination of $K(T_{photo})$, not for the MFT.

The cumulative nucleus concentration is described as (Vali, 1971; Murray et al., 2012):

$$K(T_{photo}) = -\frac{\ln(1-f_{ice})}{V} * d \qquad (2)$$

With $f_{ice}$ being the frozen droplet fraction, $V$ the droplet volume (8.2 pL for 25 µm diameter), and $d$ the dilution factor.

$$f_{ice} = \frac{n_{frozen}}{n_{total}} \qquad (3)$$

With $n_{total}$ being the total number of droplets and $n_{frozen}$ the number of frozen droplets.

The cumulative nucleus concentrations are given over the whole temperature range, further, the concentration at -34 °C was used to compare different samples. Since we have never observed homogeneous freezing of ultrapure water at temperatures of -34 °C and higher with our setup, we attribute these values purely to heterogeneous freezing events.

## 2.5. FTIR-spectroscopy

FTIR (Fourier-transform-infrared) spectroscopic measurements were conducted with a Vertex 80v (Bruker, Germany) containing an MCT (mercury cadmium telluride) detector cooled with liquid nitrogen. The optical bank was evacuated (2.6 hPa) and had a GladiATR™ single reflection ATR accessory unit (Pike, USA). The ATR unit contained a diamond crystal as total reflection window. OPUS 6.5 software was used for evaluation and instrument control. For each measurement, 128 scans were accumulated at a resolution of 0.5 cm$^{-1}$. The crystal surface was flushed with dry nitrogen to prevent humidity from interfering with the measurements.

All three extracts of TBA as well as birch pollen washing water were measured at the same conditions by preparing a thin liquid layer of the extract and evaporating the contained water with a fan. The temperature on the surface of the crystal during evaporation was always below 35 °C. This process was repeated until the dried residues of approx. 20 µL of the sample had been applied. IR spectra of all other extracts can be found in the supporting information (see Figure S1-3).

## 3 Results

### 3.1 Freezing temperature and ice nuclei concentration

All 30 extracts of birch trees were ice nucleation active (Figure 2). The highest variation in mean freezing temperature (MFT) was found for the extracts from the leaves, which showed the highest (TBC-L -15.6 °C) and lowest (TBI-L -31.3 °C) MFT amongst all analysed samples (Figure 2). Of the ten birch trees, the leaves of only five trees (TBC-L, TBD-L, TBF-L, TBG-L, VB-L) showed freezing temperatures close to the birch pollen line (-17.1 °C see Figure 2). Those samples froze between -15.6 °C (TBC-L) and -19.3 °C (TBD-L and VB-L). The remainder of the analysed leaf extracts froze at temperatures of -25.4 °C and below.

All primary wood extracts were ice nucleation active, with most MFT values between -17.5 °C (TBE-P) and -22.6 °C (TBI-P). Further, two samples froze at -25.4 °C (TBH-P, TBD-P). For most secondary wood extracts, we found slightly higher MFTs than for the primary wood samples. The values ranged from -17.2 °C (TBB-S) to -22.8 °C (TBH-S). The MFTs of the majority of the wood samples were close to the birch pollen line (-17.1 °C, see Figure 2).

However, it should be noted that a direct comparison between the different analysed samples and birch pollen washing water is not straightforward, as the freezing behaviour of the washing water is highly dependent on its concentration. We observed freezing events for birch pollen washing water and its dilutions from -15 °C down to temperatures below -35 °C, which marks the same temperature regime as the analysed birch samples exhibited freezing events. This effect is illustrated in Figure 3, showing the freezing curves of pure birch pollen washing water and two dilutions, as well as their MFT and the corresponding standard deviation.

We compared the heterogeneous freezing regime of our samples (all freezing events down to -35 °C as defined in the data analysis section) to the heterogeneous freezing regime of pure washing water as well as to the two dilutions (which correspond to samples with $10^8$ INP per mg and $10^6$ INP per mg) using the Wilcoxon–Mann–Whitney test. This is a non-parametric test which analyses if the median of the distribution function of two populations can be differentiated (DePuy et al. 2005). The used n-values, which are the same as the number of droplets frozen heterogeneously during analysis of a sample ranged between 28 and 118 for the correlated samples and between 92 and 135 for the standards. The n-values and gained data can be found in Table 2. It also gives the calculated p-values as results. The p-values are an indicator for the significance of the gained results. Calculated p-values above the set significance level α (0.1 %) indicate that the null hypothesis cannot be rejected. This means that no significant differences can be found between two distributions within the set level of significance. Similarities were shown by the test for the pure washing water with TBB-S and TBD-L. These two samples belong to the samples tested with the highest activity, which are similar to pure birch pollen washing water. With the dilution equivalent to $10^8$ INP per mg, the test shows similarities for both TBA-S and TBA-S2, and with the dilution equivalent to $10^6$ INP per mg the test shows similarities for TBD-P, TBH-P, TBH-S, and TBI-P. The latter are among the samples tested with the lowest activity and these samples match a strong dilution of birch pollen washing water (approx. 1:10,000) with only weak activity left (see Figure 3).

The cumulative nucleus concentration $K(T_{Photo})$ showed a trend similar to the MFT (as depicted for -34 °C in Figure 2, and for all temperatures above -35 °C in Figure 4). Leaf extracts mostly exhibited cumulative nucleus concentration at -34 °C between $2.8*10^6$ mg$^{-1}$ (TBH-L) and $5.0*10^9$ mg$^{-1}$ (VB-L), with two outliers exhibiting $4.6*10^5$ mg$^{-1}$ (TBI-L) and $6.7*10^4$ mg$^{-1}$ (TBE-L). However, these two outliers with the low INM concentration were the two leaf samples exhibiting the lowest MFT values (TBE-L -30.4 °C, TBI -31.3 °C). This indicates that the unusually low MFTs are a result of low concentrations of INM in the sample. Leaf extracts, which exhibited the highest variation in MFT also exhibited the highest variation in INM concentration (see Figure 2, Figure 4, and Figure 5).

The dotted line in the lower panel refers to the K(-34 °C) value of birch pollen washing water ($1.3*10^{10}$ mg$^{-1}$). Presented data shows that the samples with the highest K(-34 °C) values (TBB-S, and all samples from the Viennese birch) contain similar amounts of INP per mg extracted sample. For primary wood extracts, most values for K(-34 °C) ranged between $1.0*10^6$ mg$^{-1}$ (TBD-P) and $6.1*10^9$ mg$^{-1}$ (VB-P). Secondary wood extracts again exhibited the least variation, which can be seen best in Figure 4 and Figure 5. Their cumulative nucleus concentrations at -34 °C ranged from $4.6*10^7$ (TBD-S) to $4.6*10^9$ mg$^{-1}$ (VB-S, TBB-S). Figure 4 shows that this decreased variation compared to the other samples is not just true for the cumulative nucleus concentration at -34 °C, but over the whole temperature regime.

The averages of MFT and cumulative nucleus concentration (Figure 2) show a similar trend. Leaves exhibit lowest freezing temperature and cumulative concentration, followed by primary wood, and secondary wood exhibit highest values in both categories. This points towards a relationship between concentration and freezing temperature as it has already been observed for the birch pollen extracts.

To examine the INP distribution within a tree, a second branch of TBA was prepared and measured according to the described protocol. Resulting data are presented in Figure 2 and marked with a 2 (TBA-L2, TBA-P2, and TBA-S2). Primary and secondary wood extracts are well in line regarding their freezing temperatures (TBA-P -20.4 °C, TBA-P2 -19.8 °C; TBA-S -17.8 °C, TBA-S2 -16.7 °C), however, the primary wood from the second analysed branch contained higher INP concentrations (TBA P $2.2*10^8$ mg$^{-1}$, TBA-P2 $1.5*10^9$ mg$^{-1}$; TBA-S $2.4*10^8$ mg$^{-1}$, TBA-S2 $3.7*10^8$ mg$^{-1}$). Leaves varied in their freezing temperatures and cumulative nucleus concentrations (TBA-L -25.3 °C and $3.5*10^7$ mg$^{-1}$, TBA-L2 -21.8 and $1.0*10^8$ mg$^{-1}$).

Further, we analysed the relationship of the extractable INM concentration and the extractable total mass. The total extractable mass (given as dry mass in Figure 5) describes the weight of the dry residue of a filtered extract in mg/mL. It was highest for leaf extracts and lowest for the secondary wood extracts. As in the other attributes, leaf extracts exhibited the

highest variations with dry masses ranging from 11 (TBE-L) to 19 mg/mL (TBG-L), followed by primary wood extracts ranging from 7 (VB-S) to13 mg/mL (TBG-P). Dry masses of the secondary wood extracts ranged from 6 (TBD-S, TBE-S. TBF-S, VB-S) to 11 mg/mL (TBH-S). The secondary wood samples tended to exhibit highest concentrations of INM per mg sample mass, they also had the highest ratio of INM compared to dry mass (see Figure 5). The lowest ratio was found for the leaf extracts.

While our results show that all analysed birch trees were ice nucleation active, we also found that the trees themselves vary in their activity if compared to each other. We found lowest concentrations of INM (if all samples are regarded) for TBD, TBH, and TBI (see Figure 2 and Figure 4), all of which were growing along a riverbank with no traffic next to the trees. Only one tree with these growing conditions was found to exhibit high INM concentrations (TBC). Highest concentrations were found in the samples of the Viennese birch, located in a small park in Vienna, surrounded by heavy traffic. We found that trees, which were growing in close proximity to each other (see Figure 1), often exhibited comparable INA. This is especially true for TBA and TBB, as well as TBH and TBI. TBE and TBF match each other well except for the INA of the analysed leaves. TBC and TBD however acted significantly different if compared to each other, with TBD showing decreased INM concentrations.

### 3.2. Heat Treatment

To analyse the similarities to birch pollen washing water, all three extracts of TBA were treated at 100 °C following the protocol introduced by Pummer et al (2012). Therefore, 100 µl of each extract were applied on a clean glass slide and put in an oven set to 100 °C. After an hour the dry residues were resuspended in 100 µl of ultrapure water each and analysed for INA. The results of this experiment are given in Figure 6 as MFT and K(-34 °C) values. The corresponding values of the untreated TBA extracts are plotted for comparison. We find no major changes in the mean freezing temperatures (TBA-L -25.4 °C, TBA-L treated -26.1 °C; TBA-P -20.4 °C, TBA-P treated -20.9 °C; TBA-S -17.8 °C, TBA-S treated -18.2 °C) or K(-34 °C) values (TBA-L $3.5*10^7$ mg$^{-1}$, TBA-L treated $4.1*10^7$ mg$^{-1}$; TBA-P $2.2*10^8$ mg$^{-1}$, TBA-P treated $1.5*10^8$ mg$^{-1}$; TBA-S $2.4*10^8$ mg$^{-1}$, TBA-S treated $1.8*10^8$ mg$^{-1}$).

### 3.3. FT-IR-spectroscopy

FT-IR-spectroscopy was used to examine similarities in chemical composition between the extracts of TBA (leaves, primary and secondary wood) and aqueous birch pollen extract. The normalized FT-IR-spectra are shown in Figure 7. Table 3 contains assignments for the band positions. On the left side of the spectrum, there is a broad band with a maximum at approx. 3300 cm$^{-1}$ typical for NH and OH stretching vibrations, and further, a bisected band with maxima at 2940 cm$^{-1}$ and 2890 cm$^{-1}$, which can be assigned to aliphatic CH stretching vibrations. All four spectra show a weak shoulder at approximately 2700 cm$^{-1}$ which is linked to OH stretching vibrations. On the low-frequency side (1800 to 750 cm$^{-1}$) we find a broad array of bands. We assigned 19 maxima. Several of these bands are typical for saccharides as well as for xylan. We also found bands in all three typical amid regions. All three regions are consistent with other biomolecules (e.g. polyketides) as well; therefore the presence of peptides is not entirely clear. The spectra of the different extracts of TBA (Figure 7) show a strong resemblance to each other, but we find three main differences. (a)The intensity at 1510 cm$^{-1}$: while the band is strongly visible in the spectrum of secondary wood extracts, it is much less pronounced in the spectra of primary wood and leaf extracts. (b)The band at 1070 cm$^{-1}$ is strongest visible for the leaf extract, where it nearly swallows its neighbour at 1110 cm$^{-1}$, while it is only present as a slight shoulder for the wood extracts. (c)The region of 920 cm$^{-1}$ and below increases in intensity from leaf extract over primary wood extract to secondary wood extract.

Comparing the birch pollen washing water to the TBA extracts, we see an enhancement of the low-frequency side of the spectrum. We find all maxima present in the pollen washing water spectrum also in the other extracts: However, some bands, which are clearly pronounced in the pollen spectrum, are only very weak shoulders in the TBA extracts spectra (1350, 1300,

1270, 1200, 1140, 810, and 770 cm$^{-1}$). Furthermore, we find the maxima of the two most pronounced bands (3300 and 1050 cm$^{-1}$ given for the TBA extracts) to be shifted slightly by approx. 25 cm$^{-1}$. The spectra of all analysed samples are given in the supplementary Figures S1-3. They show the same features as the spectra given for TBA, with varying intensity ratios.

## 4 Discussion

We examined the INA of samples from ten different birch trees (*Betula spp.*) to extend the knowledge on their freezing behaviour. Samples were taken from nine birch trees in Tyrol, Austria, and from one tree in a small urban park in Vienna, Austria. Filtered aqueous extracts of 30 samples of leaves, primary wood, and secondary wood were analysed for INA using VODCA (Vienna Optical Droplet Crystallization Analyser), an emulsion technique. All of the samples from milled birch branches contained INM in the submicron size range. Such INM were previously found in other biological material

including fungi and leaf litter (Schnell and Vali, 1973; Fröhlich-Nowoisky et al., 2015; O'Sullivan et al., 2015), as well as birch pollen (Pummer et al., 2012, 2015). Our results extend these previous observations and demonstrate that aboveground material from the birch tree (and not just the pollen) can produce INM.

Several studies have found that organic components can increase the INA of soil and dust (Conen et al., 2011; O'Sullivan et
al., 2014, 2016; Tobo et al., 2014; Hill et al., 2016). Such organic components could be provided by INM released by birch trees, which could stick to inactive particles, and thus enhance their INA. Cracks and wounds on the surface could allow the INP to be washed of the surface of twigs and leaves into the soil. This marks a potential to influence the INA of mineral dust and soil particles and act as INP in the atmosphere. Huffman et al (2013) observed increased INP concentrations after rain events related to a burst in concentrations of biological particles. Eventually, INM released from plants such as birch play an
important role in this process. Further studies on possible release pathways of the INP from birches into the surrounding environment are necessary to quantify such effects.

The freezing temperature observed for the aqueous birch pollen extract (-17.1 °C see Figure 2), is in line with values reported in the literature for aqueous birch pollen extracts (reported freezing events are generally between -15 and -23 °C
(Diehl et al., 2001; Pummer et al., 2012; Augustin et al., 2013; O'Sullivan et al., 2015)). Interestingly, most of our samples froze in the same temperature range between -15 °C and -23 °C. Half of the leaves (TBC-L, TBD-L, TBF-L, TBG-L, and VB), eight out of ten primary wood samples (TBA-P, TBB-P, TBC-P, TBE-P, TBF-P, TBG-P, TBI-P, and TBV-P) and all secondary wood samples exhibited a mean freezing temperature in this temperature window. If we broaden the temperature window by including dilutions of birch pollen washing water, we observe freezing events happening down to -35 °C. Nearly
all freezing events recorded for the presented samples freeze in this temperature window between -15 and -35 °C. Leaves only occasionally show higher freezing events (see Figure 2). The Wilcoxon–Mann–Whitney test shows similarities between more than a quarter of our samples and birch pollen washing water as well as the respective dilutions (see Table 2). These samples showed some of the highest and lowest MFT values measured (e.g. TBB-S, the best secondary wood sample matches pure washing water, while the lowest, TBH-S matches the dilution of birch pollen washing water, that is equivalent
to $10^6$ INP per mg). As we were able to match some of the most and least active samples, we conclude that there is a correlation for each sample between concentration of INP and freezing temperature relating to the birch pollen washing water. Moreover, we observed heat resistance at 100 °C, similar to the results of Pummer et al. (2012). This indicates a resemblance between the INM from pollen and those found in the extracts of leaves, primary wood, and secondary wood. The data shows that the average freezing temperatures of secondary wood, primary wood, and leaves differ slightly. These
differences however follow the same pattern as the INM concentration. Therefore we assume this to be a concentration effect. Based on these results, we hypothesize that the INM in birch, which are found inleaves, primary wood, and secondary

wood behave similar and can be statistically related to the INM found in birch pollen. This means that INM from birch trees are not just relevant during the pollen season but over a longer period of time, possibly even over the whole year. It is important to conduct further research on the seasonal dependency of the production of INM of birch trees.

We observed a high variability of INM in leaves. Even for leaves of two branches of the same tree, we found differences in their freezing temperatures. Only five out of ten samples froze at similar temperatures as the INM from birch pollen. The high variability could be explained by external impacts, as leaves are easily influenced by their growing conditions. Leaves growing in the shade exhibit reduced dry masses and nitrogen content (Eichelmann et al., 2005). Also their hydrological conductivity is impacted by radiation (Sellin et al., 2011). Further, the growing site next to a river typically leads to enhanced

water availability, which can cause increased leaf conductivity and transpiration rate in the lower crown foliage of trees (Sellin and Kupper, 2007).

Birches are native through most of Europe, even up to central Siberia and are capable of growing in boreal regions and high altitudes (Beck et al., 2016). Due to climate change and their resistance against cold climate, some birches can even be found

above the tree line (Truong et al., 2007). Due to this vast distribution area, the growing conditions of birches may vary greatly. We speculate that environmental conditions may influence the production and release of INM from birch. Many environmental factors can affect the plant physiology and growth as e.g. humidity (Sellin et al., 2013), atmospheric ozone (Maurer and Matyssek, 1997; Harmens et al., 2017), $CO_2$ (Rey and Jarvis, 1998; Kuokkanen et al., 2001), $NO_x$ and $SO_2$ (Freer-Smith, 1985; Martin et al., 1988), as well as exposure to light and its wavelength (Eichelmann et al., 2005; Sellin et

al., 2011). All of these factors might also influence the INM production of birch trees. The extracts of branches showed a systematic distribution of INM that could relate in part to their growth environment. Wood samples of the birches TBD, TBH, and TBI (see Figure 2), which all three were growing next to a river, froze at lower temperatures than the other birch samples. Moreover, there was a tendency for birch trees located near roads (TBA, TBB, TBE, TBF, and VB grew directly next to roads) to be associated with increased INA. TBE and TBF grew next to a road and a river, but showed comparable

INM concentrations to the other road side birches. If the INA of birches is based on a stress or defence mechanism, this could be due to stress caused by the exhaust of traffic, e.g. $NO_x$, which is an important pollutant released by traffic, (Franco et al., 2013) and has the potential to harm plants, but can also be absorbed by many plants and used as a nitrogen source (Allen Jr, 1990). Other than roads and rivers in close proximity, the tree altitude was not correlated to INA. Future investigations of birch trees located across different altitudes, roads, settlements, and forests are warranted.

Some investigations on birch stands showed a dry weight of 2 to 25 t per ha for twigs and 1 to 8 t per ha for leaves (Johansson, 1999; Uri et al., 2007). This leads to estimated INP concentrations on the order of $10^{16}$ to $10^{19}$ per ha for twigs and $10^{14}$ to $10^{18}$ per ha for leaves. Plant debris can be an important constituent of ambient particulate matter (Matthias-Maser and Jaenicke, 1995; Andreae, 2007; Winiwarter et al., 2009). However, the underlying processes of the release of plant debris in the atmosphere is not fully understood, making predictions of their atmospheric impact hard (Andreae, 2007;

Winiwarter et al., 2009). Sánchez-Ochoa and colleagues analysed atmospheric aerosols collected at various background sites in Europe and used cellulose as a proxy for plant debris. They found biannual average concentrations of 33.4 to 363 ng per $m^3$ air (Sánchez-Ochoa et al., 2007). Especially the leaves of birch trees could be an important source for INP as it is shed and produces annually. Decaying leaf litter is known to be a good source of INP (R.C. Schnell and Vali, 1973). Conen et al. (2016, 2017) showed that air masses passing over land can be enriched with INP derived from such leaf litter. Collectively,

these studies underscore the importance of plants as sources of INP.

Since all analysed materials are of natural origin, we cannot rule out that some contamination could play a role in the INA of our extracts. Some bacteria have been found to act as INP (as e.g. *Pseudomonas syringae* (Maki et al., 1974)), however,

these bacteria are typically in the size range > 1 µm (Monier and Lindow, 2003) and therefore easily filtered with the 0.2 µm syringe filter. Further, some lichen are known to be INA (Kieft, 1988), and some microorganisms release their small contained INP in the aqueous phase as e.g. *Mortierella alpina* (Fröhlich-Nowoisky et al., 2015), which cannot be filtered with used methods. However, most known ice nucleation active lichens and microorganisms as well as released INP

typically freeze at significantly higher temperatures (above -10 °C (Maki et al., 1974; Kieft, 1988; Pouleur et al., 1992; Murray et al., 2012; Fröhlich-Nowoisky et al., 2015) than the freezing temperatures observed for our samples, with very little exceptions (Iannone et al., 2011). As the highest onset temperature observed in our measurements was -14.1 °C (TBC-L), and the onset temperature of birch pollen washing water was quite close to this value (-15.1 °C), and heat treatment did not affect the extracts of TBA, we do not suspect significant contamination of our samples. However, the INA of birches,

especially if growing close to a road or in urban regions, could be affected by soot and other anthropogenic emissions, as soot can act as INP (DeMott, 1990; Murray et al., 2012)

The measured FTIR spectra indicate that the birch extracts are chemically similar to each other, and to pure birch wood. As plants do not only contain polysaccharides but several soluble carbohydrates (Magel et al., 2000), we assume those

substances to play an important role in the chemical composition of our extracts. Fitting to this assumption, most of the bands found in our spectra could be assigned to carbohydrates and polysaccharides. Presented spectroscopic data matched the literature well (Chen et al., 2010; Pummer et al., 2013; Dreischmeier et al., 2017), however intensity ratios varied. IR spectra from birch pollen and TBA extracts (Figure 7) show strong similarities to the spectrum of milled birch wood shown by Chen et al. (Chen et al., 2010) (measured in KBr pellets). In particular, the range between 1150 and 1300 cm$^{-1}$, i.e.

especially the band at 1270 cm$^{-1}$, was strongly enhanced compared to our spectra. Also, the band at 1510 cm$^{-1}$ was very intense in the pure wood spectrum compared to the extracts. Both bands are representative for lignin, a main substance in wood that is only weakly soluble in water. Since the remaining weak bands can be assigned to other structural elements, our extracts likely did not contain any lignin. Other than the lignin bands, our aqueous extracts show very similar spectroscopic features compared to the pure wood samples. These similarities between the spectra of extracts and the spectrum of pure

wood indicate that our extract method retrieves the majority of components, leading to a similar distribution of bands, with differing intensities due to differences in concentration. The IR spectrum of birch pollen washing water (Figure 7) is in well agreement with the literature data (Pummer et al., 2013; Dreischmeier et al., 2017). The extracts of the different TBA samples (leaves, primary wood, and secondary wood) exhibit similar spectra with no major differences. The birch spectra of birch pollen washing water and the different wood extracts match well, showing very similar maxima with mostly minor

differences in intensity ratios.

In the FTIR spectroscopy data, we found strong similarities between birch pollen washing water and the different aqueous extracts from the TBA samples. Further comparison with whole pollen grains, as well as with pure wood, as found in literature, shows strong similarities in the spectroscopic features of our different birch samples. As not just the band position,

but also the intensity ratios are agreeable with each other, we assume this to indicate that we are able to extract the major components found in wood with our extraction method and that the pollen and wood samples extracts exhibit chemical similarities to a certain extent.

Only little INP are known to trigger freezing above -10 °C, which are typically biological substances such as bacteria

(Murray et al., 2012). Below -10 °C, birch pollen belong to the group of highest freezing temperatures, with onset higher than most mineral dusts, ash and soot samples (Murray et al., 2012). The vast majority of atmospheric INP and INP retrieved from precipitation samples exhibit freezing temperatures below -10 °C (DeMott et al., 2010; Petters and Wright, 2015). The identity of the INP released from birches is still unclear. Pummer et al. (2013) showed that proteins, saccharides, and lipids

are easily extracted aqueously from birch pollen. While Pummer et al. (2012) and Dreischmeier et al. (2017) speculate that the responsible molecules are carbohydrates, Tong et al. (2015) attributes the highest INA to extracted proteins. Hiranuma et al. (2015) showed that cellulose, a polysaccharide which is ubiquitous in plants, exhibits INA in the same temperature range With our spectroscopic data, we found strong indicators for saccharides being present, including prominent bands which could be associated with cellulose. Further, we found bands in the most prominent protein regions, though those could be assigned to other molecule groups.

**5 Conclusion**

Ten different birch trees (*Betula spp.*) were examined. Filtered aqueous extracts of 30 samples of leaves, primary wood, and secondary wood were analysed for INA using VODCA, an emulsion technique. All samples contained ice nuclei in the submicron size range. Concentrations of ice nuclei ranged from $6.7*10^4 – 6.1*10^9$ per mg. Mean freezing temperatures varied between -15.6 °C and -25.4 °C (excluding three samples that exhibited lower freezing temperatures). Comparing the freezing behaviour of our samples to birch pollen washing water and two dilutions (1:100 and 1:10,000) using the Wilcoxon–Mann–Whitney test ,we found statistical correlations for more than a quarter of our samples and birch pollen washing water, indicating a relationship between the INM of wood, leaves and pollen. As we were able to match some of the most and least active samples, we conclude that given the right dilution of birch pollen washing water, all samples could be matched. The majority of the samples showed freezing temperatures close to those of birch pollen extract, indicating a relationship between the INM of wood, leaves and pollen. Extracts derived from secondary wood showed the highest concentrations of INM and the highest freezing temperatures. Extracts from the leaves exhibited the highest variation in INM and freezing temperatures. Infrared spectra of the extracts suggest that the aqueous extracts of birch materials tested showed similarities to aqueous extracts of birch pollen. Our results suggest that there might be linkages between INA, growing site, and condition of the birch tree, with streets exhibiting a positive influence and rivers tending to exhibit a negative influence on INA. Field and laboratory studies are needed to examine how much ice nucleation active material can be expected per surface area of a tree and which quantity of this material can be aerosolized. A broader selection of samples is also needed to further examine differences between different trees and an influence of growing site and season.

**Acknowledgments**

The authors would like to thank the FWF (Austrian Science Fund, Project No. P 26040) and the FFG (Austrian Research Promotion Agency, Project No. 850689) for funding.

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

**Table 1: Further information on the sampled birches, with sample name, circumference, GPS waypoints, altitude of the growing site and a further description thereof:**

| ID of birch tree | GPS waypoints | GPS altitude | Circumference at 1 m height | Location description |
|---|---|---|---|---|
| TBA | 47.214241, 10.798765 | 799 m | 113 cm | Roadside birch in the valley |
| TBB | 47.221615, 10.829835 | 799 m | 54 cm | Roadside birch in the valley |
| TBC | 47.186231, 10.908341 | 851 m | 75 cm | River side birch in the valley |
| TBD | 47.185387, 10.909587 | 851 m | 35 cm | River side birch in the valley |
| TBE | 46.973163, 11.010921 | 1343 m | 96 cm | River side birch in Sölden next to a road with little traffic |
| TBF | 46.974588, 11.011463 | 1343 m | 61 cm | River side birch in Sölden next to a road with little traffic |
| TBG | 46.878959, 11.024441 | 1925 m | 67 cm | Timberline birch, the last birch and one of the last trees in general we encountered on our way up |
| TBH | 46.873275, 11.026616 | 1883 m | 36 cm | Riverside birch in Obergurgl close to the timberline |
| TBI | 46.873279, 11.026736 | 1883 m | 59 cm | Riverside birch in Obergurgl close to the timberline |
| VB | 48.197796, 16.352189 | 195 m | 86 cm | Located in the centre of a small park in Vienna, which is surrounded by heavy traffic |

**Table 2: The results for the Wilcoxon–Mann–Whitney test. All given samples were shown to match birch pollen washing water or a dilution thereof. Pure birch pollen washing water is marked with pure, the 1:100 dilution equivalent to $10^8$ INP per mg is marked with 1:100, the 1:10,000 dilution equivalent to $10^6$ INP per mg is marked with 1:10,000. n represents the number of data points used for comparison for each sample[a]. The used significance level $\alpha$[b] was 0.1 %:**

| Sample | n | Birch pollen washing water concentration | p-value[c] |
|---|---|---|---|
| TBB-S | 50 | Pure | $6.6*10^{-3}$ |
| TBD-L | 70 | Pure | $2.8*10^{-1}$ |
| TBA-S | 48 | 1:100 | $1.3*10^{-3}$ |
| TBA-S2 | 118 | 1:100 | $2.7*10^{-1}$ |
| TBD-P | 28 | 1:10,000 | $8.3*10^{-2}$ |
| TBH-P | 117 | 1:10,000 | $1.9*10^{-3}$ |
| TBH-S | 102 | 1:10,000 | $1.3*10^{-1}$ |
| TBI-P | 43 | 1:10,000 | $6.6*10^{-2}$ |

5    [a]n is equivalent to the number of droplets frozen homogeneously. n-values of the used standards were: (pure washing water)=96, n(1:100)=135, n(1:10,000)=92.

[b]The significance level marks the probability of falsely assuming two populations to differ in their distribution.

[c]The p-value indicates the significance of the result. If the p-value is higher than the used significance level, the statistics indicate no differing between two distributions.

**Band assignment of the IR spectra of TBA extracts (leaves, primary wood, and secondary wood) and birch pollen washing water (Miyazawa et al., 1956; Kačuráková et al., 2000; Schulz and Baranska, 2007; Chen et al., 2010; Pummer et al., 2013)::**

| Band wavenumber [cm$^{-1}$] | Assignment of IR spectra |
| --- | --- |
| 3300 | O-H stretch/ N-H stretch |
| 2940 | C-H stretch |
| 2890 | C-H stretch |
| 2700 | O-H stretch |
| 1720 | C=O, xylan |
| 1650 | C=O stretch, C=C, Amid I |
| 1600 | C=O stretch (lignin), C=C, Amid I, |
| 1510 | C=O stretch (lignin), Amid II, |
| 1450 | CH$_2$ deformation (lignin and xylan) |
| 1425 | Aromatic skeletal combined with C–H |
| 1350 | C-H deformation (ring) |
| 1300 | N-H C-H deformation, Amid III |
| 1270 | C=O stretch (lignin), Amid III |
| 1240 | C-O, C-N, C-N-C, C-C-O of phenolic compounds, Amid III |
| 1200 | Phosphate, C-C-O of phenolic compounds |
| 1140 | C-O-C stretching (pyronase rings), C=O stretching (aliphatic groups), Guanine, Tyrosine, Tryptophane |
| 1110 | Sugar skeletal vibration |
| 1070 | C-H stretch, C-C stretch |
| 1050 | C-H stretch, C-C stretch, Guaiacyl units (Lignin) |
| 990 | OCH$_3$ (polysaccharides) |
| 920 | C=C, cellulose P-chains, polysaccharides - β-linkage, phenolic compounds |
| 850 | C-O-C skeletal mode (polysaccharides - α-linkage, COPOC RNA, phenolic compounds |
| 810 | C=O deformation (polysaccharides), phenolic compounds |
| 770 | Phosphate stretch |

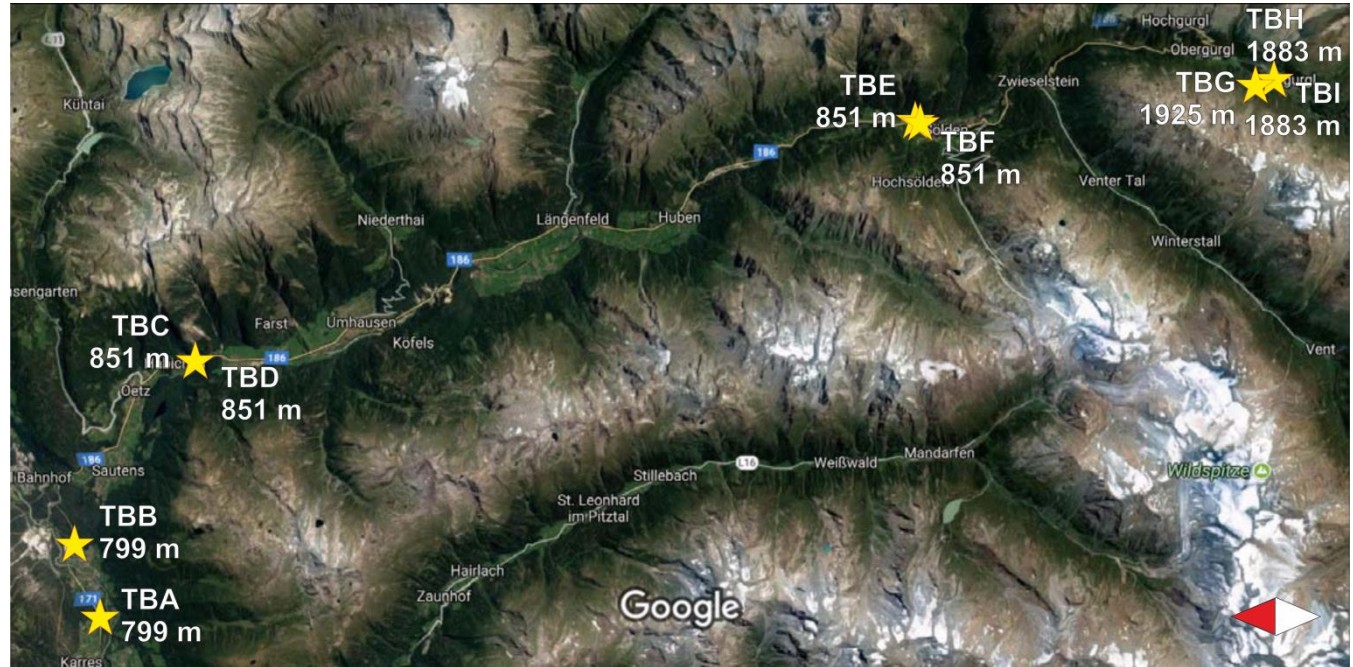

**Figure 1: Sampling sites in Tyrol along a valley with an altitudinal gradient (adapted from Google Maps, 2017). Markings for TBH and TBI, as well as for TBC and TBD completely overlap each other due to the close proximity of their growing sites.**

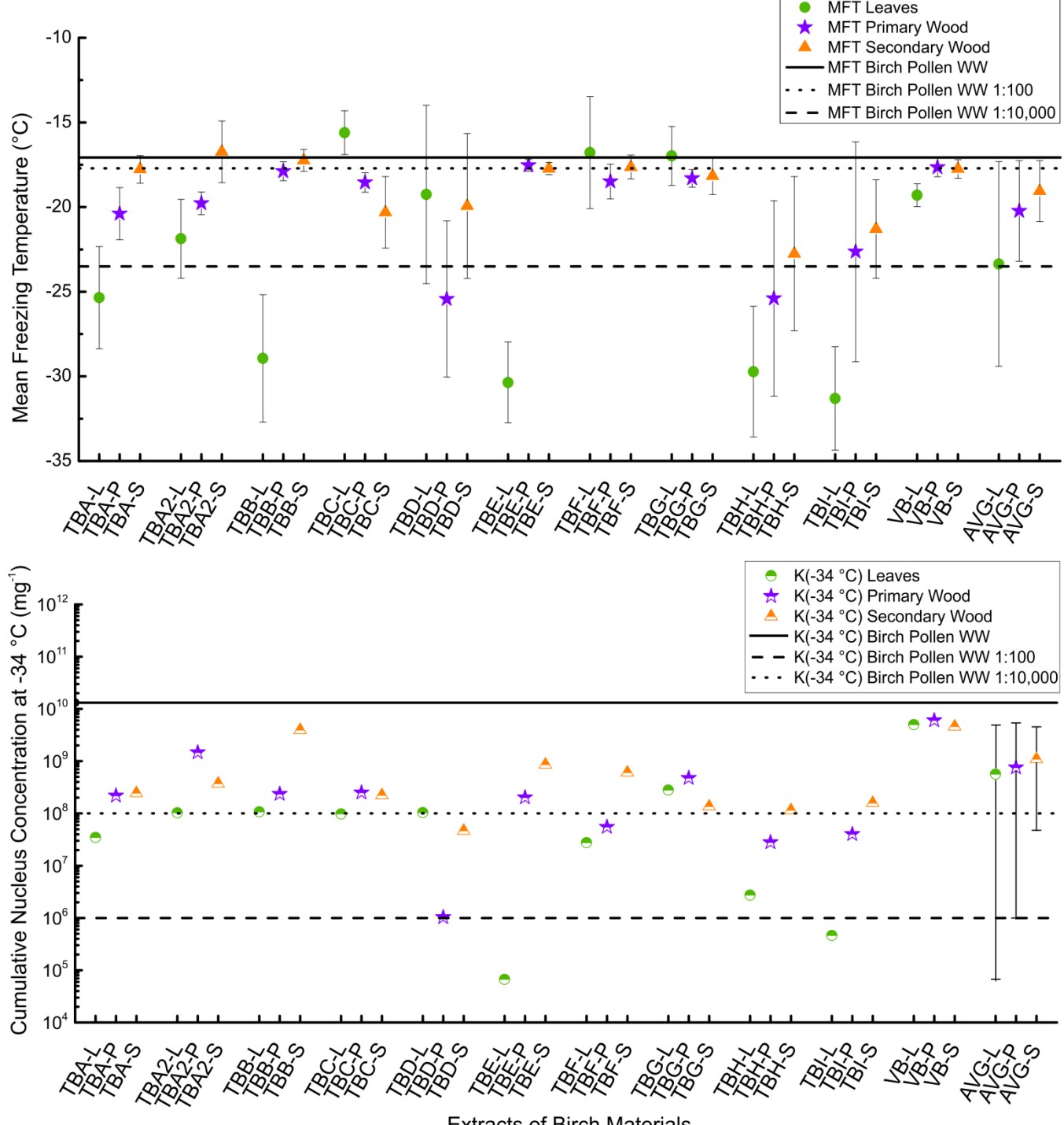

**Figure 2: Top panel: Mean freezing temperature (MFT) of the different birch samples. Leaf extracts (L) are marked with a green circle, primary wood extracts (P) with a violet stars, and secondary wood extracts (S) with an orange triangle. The solid line is the MFT of birch pollen washing water (-17.1°C with a standard deviation of ±0.5 °C (not plotted)). The dotted line represents the MFT of a dilution equivalent to $10^8$ INP per mg (-17.7 °C with a standard deviation of ±1.1 °C (not plotted)) and the dashed line refers to the MFT of a dilution equivalent to $10^6$ INP per mg (-23.5 °C with a standard deviation of ±3.6 °C (not plotted)). The last three values on the right side of the top penal represent the average of all mean freezing temperatures for leaves (AVG-L), primary wood (AVG-P) and secondary wood (AVG-S) with the corresponding standard deviation. Bottom panel: cumulative nucleus concentration at -34 °C (K(-34 °C)) of the different birch samples per mg extracted sample. Assignment of the symbols is similar to the top plot. The solid line refers to the K(-34 °C) of birch pollen washing water per mg extracted pollen ($1.3*10^{10}$ mg$^{-1}$), the dotted and dashed line refer to the dilutions from birch pollen washing water introduced in the top panel. The last three values on the right side represent the average of all K(-34 °C) values. Error bars point to the area of trust, ranging from the highest to the lowest measured values.**

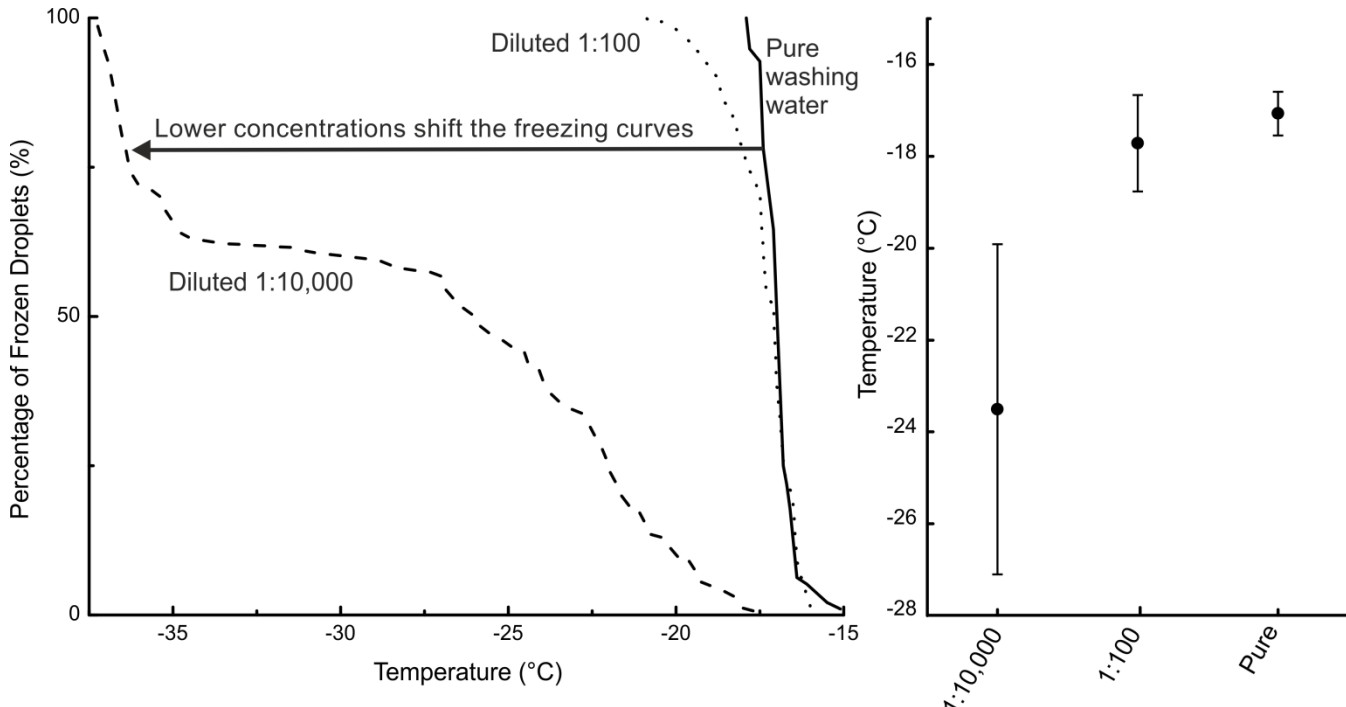

**Figure 3: Left: The freezing curves of birch washing water and two dilutions thereof (the dilution 1:100 is equivalent to $10^8$ INP per mg, the dilution 1:10,000 is equivalent to $10^6$ INP per mg) with the fraction of frozen droplets as a function of temperature. Right: the MFT with the corresponding standard deviations of the birch pollen washing water and the two analysed dilutions.**

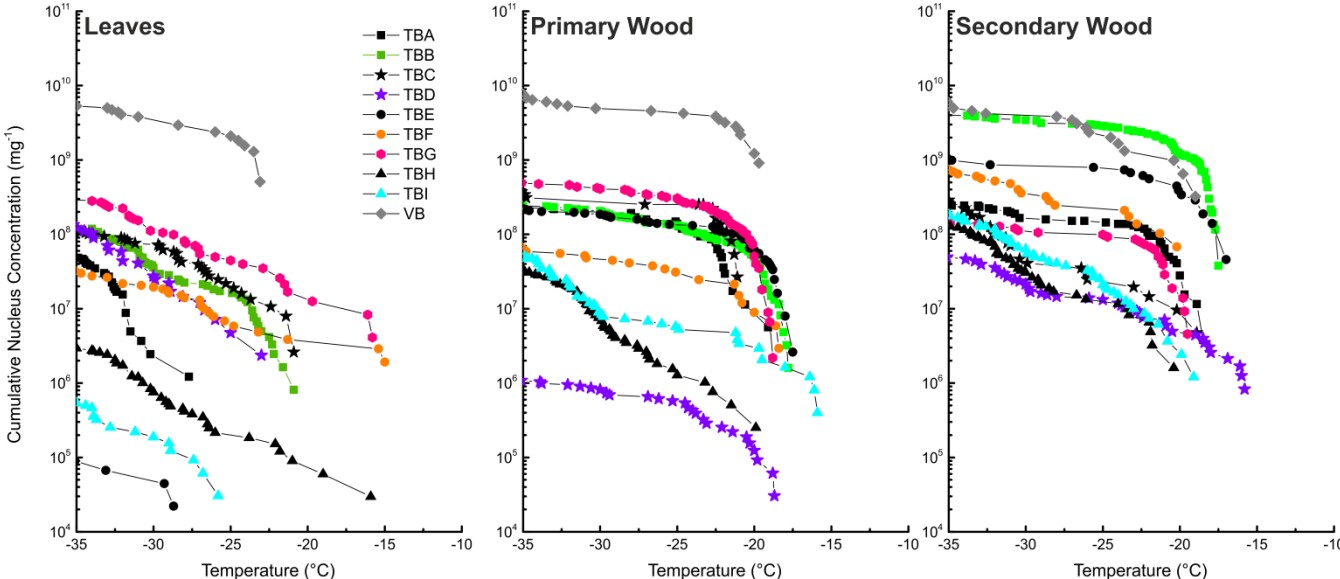

**Figure 4: Cumulative nucleus concentration as a function of temperature for leaf extracts (right), primary wood extracts (middle), and secondary wood extracts (right). The diagram is cut off at -35 °C, since we cannot contribute freezing events below this temperature to heterogeneous nucleation. The symbols used for the different data points are grouped. Birches growing in close proximity under similar conditions are marked with the same symbol (different fillings).**

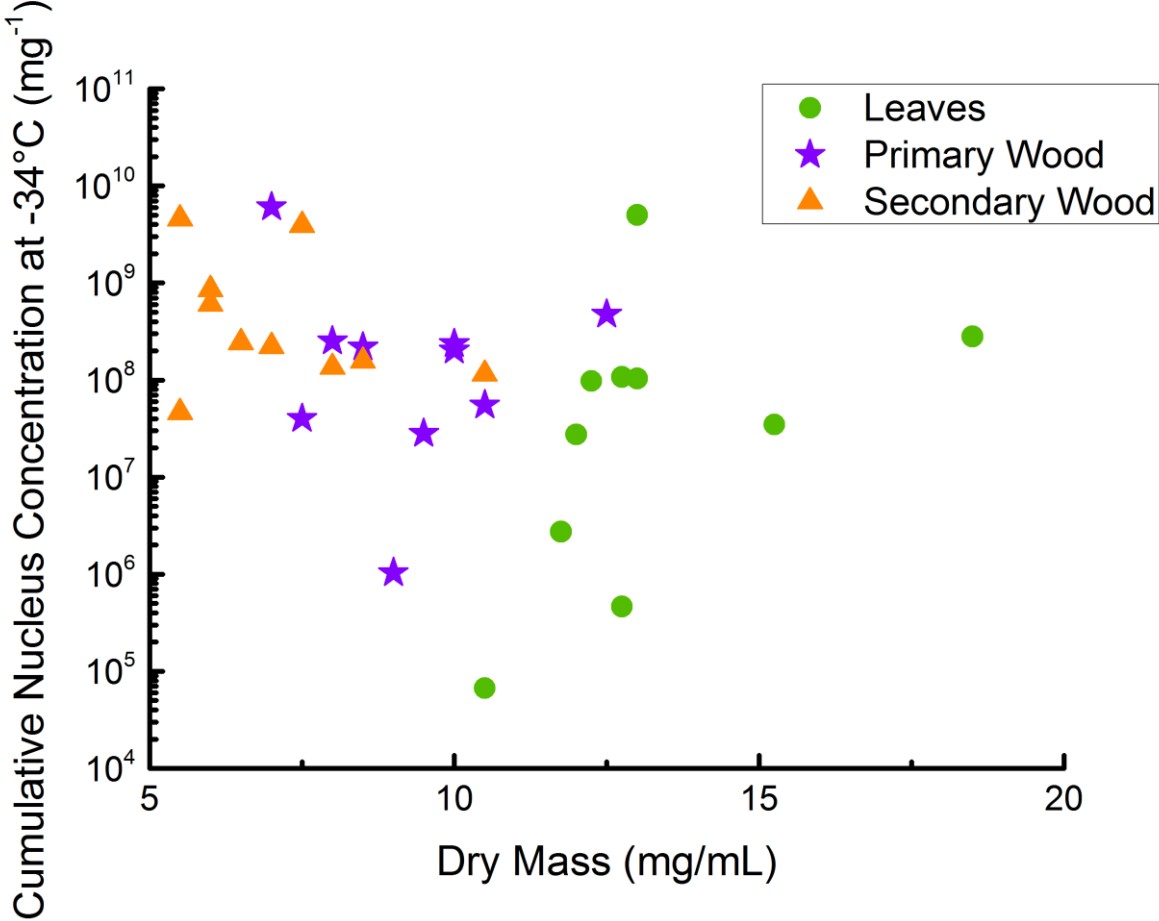

**Figure 5: Scatterplot of dry mass (dry residues of the different filtered extracts) and cumulative nucleus concentration at -34 °C per sample mass. The dry mass is the mass we were able to extract with the 50 mg/mL suspensions. The data show that secondary wood, which contained mostly the highest INM concentrations and lowest variations between different samples, also contained the lowest extractable mass. Therefore INM ratios in the extractable content of the different samples were highest in secondary wood samples.**

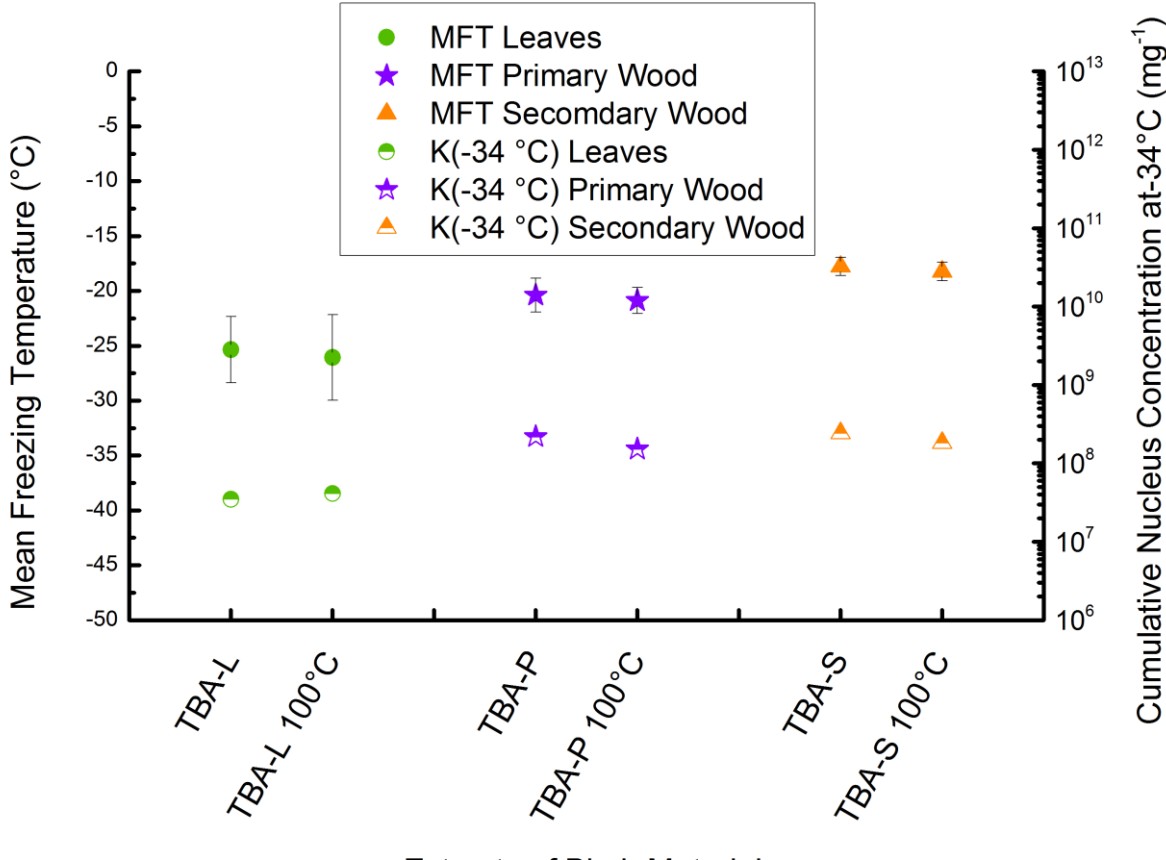

**Figure 6: Results of the heat treatment of the different TBA extracts. Leaves are marked with green circles, primary wood with violet stars and secondary wood with orange triangles. The left value belongs to the untreated sample, the right value to the sample treated with 100 °C for an hour. Filled symbols represent the mean freezing temperature and correlate with the left Y-axis, half filled symbols represent the cumulative nucleus concentration as -34 °C per mg extracted sample and correlate with the right Y-axis.**

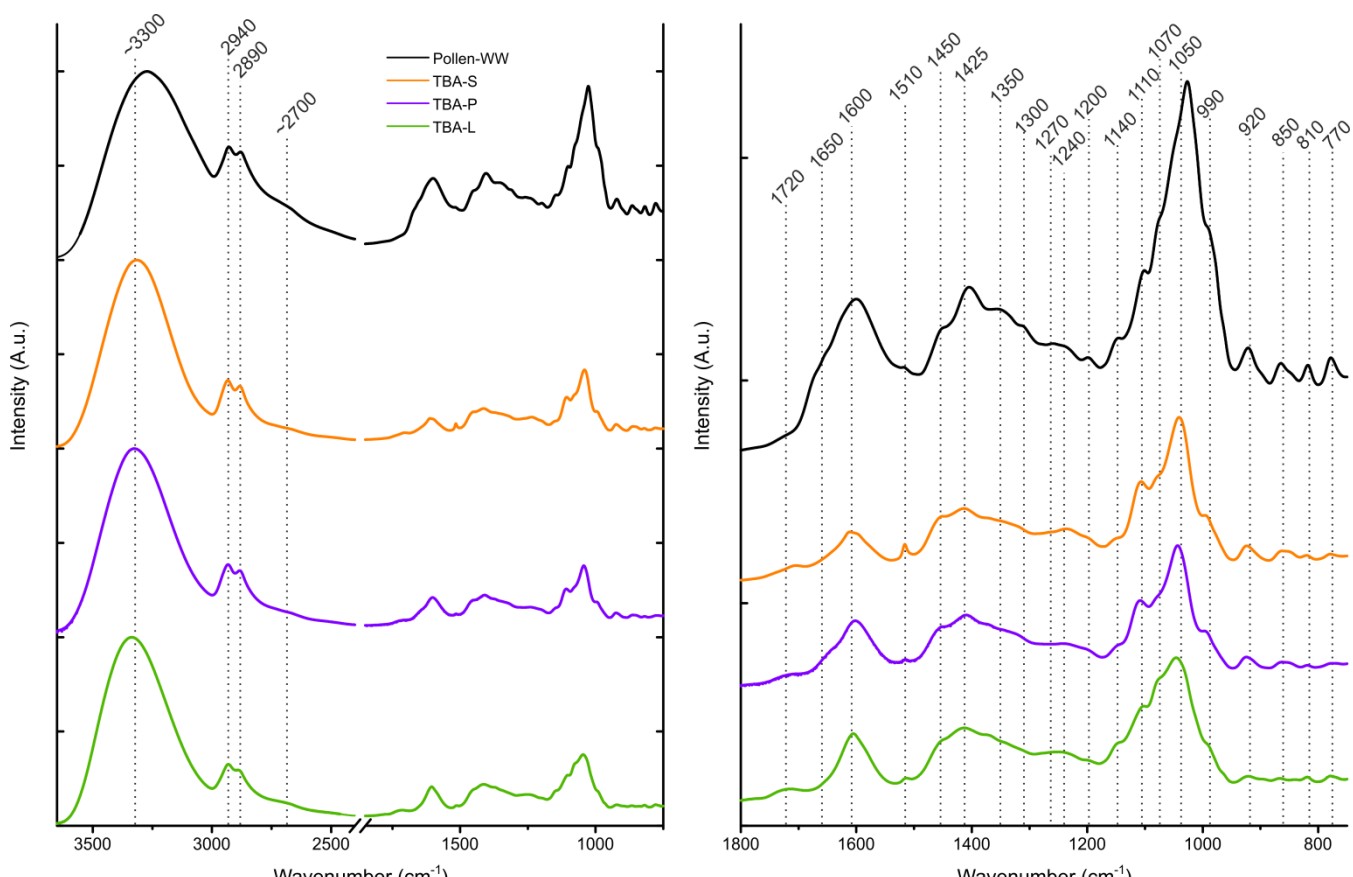

**Figure 7: FTIR spectra of the TBA extracts (leaves in green, primary wood in violet and secondary wood in orange) and birch pollen washing water (black). Left: the whole spectrum between 3650 cm[-1] and 750 cm[-1]. Right: enlarged right side of the spectrum between 1800  cm[-1] and 750 cm[-1]. Possible band assignments are given in Table 3.**

