# Peer review of "Birch leaves and branches as a source of ice-nucleating macromolecules"

_Atmospheric Chemistry and Physics, 2017_

## Referee Comment (RC1) · Anonymous Referee #1 · 12 Jan 2018

The paper presents analyses of ice nucleating particles (INPs) found in extracts of finely milled leaves, twigs, and branches from ten birch trees in Austria. It is a valuable contribution to the literature on potential sources of INPs found in the atmosphere. Its clear presentation provides for pleasant reading.

More discussion on the following issues would make the paper even stronger:

a.) Spectral analysis of extracts lead to the conclusion that birch leaves, twigs and branches contain chemical substances similar to those in birch pollen, which implies that INP in either material carry of the same sort of ice-nucleating macromolecules (INM). If so, leaf, twig, and branch INM should equally withstand denaturation at temperatures up to 445-460 K, which clearly distinguishes birch pollen INM from bacterial and fungal INM that are already denatured at much lower temperatures (Pummer et

al., 2015, https://doi.org/10.5194/acp-15-4077-2015). Did you test the heat tolerance of your samples? If so, what was the result?

b.) Another issue I would like to see addressed with regard to the nature of the INM is whether they could be a form of cellulose. This issue could be discussed with reference to the FTIR spectra in Figure 5 and also with regard to the slope of the cumulative nucleus spectra (Figure 3), as compared to similar spectra available for cellulose (e.g. Hiranuma et al., 2015, doi:10.1038/ngeo2374).

c.) In the Discussion you write that INM could be "...washed into the soil during rainfall..." (page 7, lines 29-30). Leaves and twigs are usually covered by a thin layer of wax to protect against desiccation. I wonder whether INM sitting in the tissues below the protective outer layer could be washed off. Wouldn't leaves and twigs first need to be shed and to disintegrate for INM to be washed off in larger numbers?

d.) In Section 2.1 you introduce the altitudinal gradient along which you sampled the trees. Later in the paper there seems no further reference to this gradient. Instead, you relate results to the proximity of the trees to road or river. Is altitude irrelevant for the production of INM? Could similarity in terms of INM in a particular kind of location result from a genetic proximity of the trees (i.e. seeds spreading along a road or a river)?

Minor comments

Page 2, line 9: Please be more precise. Concentrations reported by Christner et al. (2008) were quite low (at -10 °C: 4 to 490 INP/L) compared to other studies (up to 500'000 INP/L at -10 °C; Petters and Wright, 2015, dx.doi.org/10.1002/2015GL065733). What the paper by Christner et al. (2008) indeed has clearly shown was the large fraction (95%) of biological INP in the total number of INP.

Page 2, line 20: 'mechanism' seems more appropriate here than 'tool' (same in line 35).

The term "tissue" you use to denominate your samples does not seem correct to me. As I understand, you processed entire leaves and sections of twigs and branches, which you call primary and secondary wood. Branches, for example, are made up of several types of tissue (xylem, phloem, sclerenchyma, cortex, epidermis). I would find it more appropriate to not talk about "tissue" in your context but say that you analysed material from different parts of the trees (leave, twig, branch).

Trees differ in MFT and cumulative nucleus concentration in leaves. How reproducible are these values? Did you prepare and analyse, perhaps during the preparatory phase of your study, two or more samples from the same tree, i.e. did you process from one or several trees two sets of leaves or two sets of twig material?

---

## Referee Comment (RC2) · Anonymous Referee #2 · 6 Feb 2018

The manuscript submitted by Felgitsch et al. discusses work analyzing samples of leaves and wood from a number of birch trees in Austria. The authors subjected the samples to ice nucleation tests as well as to infrared and fluorescence spectroscopy. The main conclusions of the work were to report the concentration range of ice nuclei and the mean freezing temperature per sample. They also reported the spectral properties of a few selected samples to suggest that the chemical components in the samples were similar. The work showing data about ice nuclei in the birch material is interesting and could be useful to compare against other material types. The discussion was very thin and didn't make strong connections between the ice nuclei and spectroscopic analyses. The sections about optical spectral were not convincing. It wasn't obvious why infrared and fluorescence spectroscopy was used as the primary

"chemical" test for comparison here. These shown broad similarity between the few samples discussed, but it was unclear that the spectral features shown would have been expected to be different between these different portions of the same plant. To make this portion more convincing, I think it would be useful to bring additional comparison to spectra of the same components of trees in order to show either comparability or contrast.

It was also not clear what the atmospheric implications of this work would be. I understand that atmospheric relevance of small pollen particles, and to a lesser extent leaves that could fragment and disperse as aerosols in the atmosphere. The relevance of ground wood as ice nuclei was less clear. The authors attempted to make a case that the chemical components of the birch material was relatively similar across the portions of the plant and that aspects of the macromolecules could play a role in atmospheric ice nucleation. I didn't quite see the link, and I think the manuscript could be improved by more clearly discussing a direct link between the observations reported and some atmospheric implications.

Overall, the manuscript does have some interesting and novel data, and so those are worth publishing in some form. I suggest that the discussion be improved to make some of the statements a bit clearer, more defensible, and more obviously linked to ambient atmospheric processes. Additional, specific statements are listed below.

Pg 3, line20 – The authors removed "visible" contamination such as lichen. How might leaving "sub-visible" contamination affect the outcomes? I would think that removing only the obvious layer could include still significant amounts of nuclei that could still influence results. Alternatively, by taking the same sample and stripping the outer bark so that there was no possible contamination between external molecules (whether lichen, deposited pollutants, etc.) could isolate this issue. Pg 3, line27 – The drying process was continued until the weight was constant. How did the authors define "constant?" Pg5, related to Fig. 2 – Since the authors draw conclusions about the types of birch material (leaves, primary wood, etc.), it would be good to show averages + std dev

[Figure]

of each type on either the left or right within this figure. Pg 6, line33 – "pointing to the importance of polysaccharides in our extracts" This is an example of an overstatement, in my opinion. While the polysaccharides may include these specific infrared bands, fundamentally these are vibrational features of individual chemical bonds that can exist in many types of molecules. Pg 6, line 34 – "can be assigned to other biomolecules" . . . similar to the comment above. I think it would be better stated as "are consistent with" in place of "can be assigned to" Pg 7, Section 3.4 – Subtle differences in intensity of fluorescence peaks here could easily be a function of analyte concentration. How did the authors control for concentration? If the authors are suggesting that the ∼10% differences in the peak heights (e.g. of the 260 nm Ex) are due to chemical or biological differences in the sample, they should discuss how they are confident it is not just subtle dilution effects. Pg7, line 24 – "Most of our samples froze at temperatures close to the freezing temperature of birch pollen washing water." This line is a bit vague. What do the authors mean by "close to" here and "most?" Pg 8, line33 – "show strong similarities .. shown by Chen et al." Can the authors expand the discussion on this point? After looking up the spectra shown by Chen et al., I was a bit confused. I see that the Chen spectra seem to be somewhat higher in resolution, but otherwise I wasn't sure what specific points the authors were trying to extract from the comparison. Pg 8, first paragraph – How would these spectra look if you did the same with material from other tree species? Fluorescence spectra are always broad (i.e. compared to IR spectra), and then when grinding large volumes of material to be mixed into a sample for a spectrum – the analysis is obviously very homogeneous and mixed with huge numbers of types of molecules. It does not surprise me that these four sets of spectra look similar – it would surprise me if they looked very different. In contrast, I would expect the same spectra from another tree species to look very similar, so it is hard to know what this fluorescence spectra adds to the overall analysis in the manuscript. Can the authors provide comparisons to fluorescence spectra published elsewhere? Surely this has been done and is otherwise reported. Page 9, line 24 – "suggest that birch tissues tested contained chemical substances similar to birch pollen." I disagree

with the weight of this statement. I think that the results suggest that the samples may have exhibited broadly similar IR and fluorescence spectral features, but to extend the statement to say that the "chemical substances" were similar was never tested directly here. Also, the data shown in the paper suggest that spectra from different types of material from the same plant are relatively similar, but differences across plant samples are not directly shown.

Figures – In general, I would suggest using color for figures 2-4. For Fig 2, I would also put the circle/triangle/star detail into the figure legend, and not just in the caption. This would make the complex figure easier to read.

Figure 3 – How do these data compare to other atmospheric measurements using similar techniques?
* * *

---

## Author Comment (AC1) · 16 Apr 2018

The authors would like to thank the referee for the time and effort in reviewing our manuscript „Birch leaves and branches as a source of ice nucleating macromolecules"

*a.) Spectral analysis of extracts lead to the conclusion that birch leaves, twigs and branches contain chemical substances similar to those in birch pollen, which implies that INP in either material carry of the same sort of ice-nucleating macromolecules (INM). If so, leaf, twig, and branch INM should equally withstand denaturation at temperatures up to 445-460 K, which clearly distinguishes birch pollen INM from bacterial and fungal INM that are already denatured at much lower temperatures (Pummer et al., 2015, https://doi.org/10.5194/acp-15-4077-2015). Did you test the heat tolerance of your samples? If so, what was the result?*

Response: We conducted heat experiments at 100 °C, which showed no changes in the INA of the different extracts from the birch TBA. We included these results in section 3.2 (p7, 11-18):

*"To analyse the similarities to birch pollen washing water, all three extracts of TBA were treated at 100 °C following the protocol introduced by Pummer et al (2012). Therefore 100 µl of each extract were applied on a clean glass slide and put in an oven set to 100 °C. After an hour the dry residues were resuspended in 100 µl of ultrapure water each and analysed for INA. The results of this experiment are given in Figure 1 as MFT and K(-34 °C) values. The corresponding values of the untreated TBA extracts are plotted for comparison. We find no major changes in the mean freezing temperatures (TBA-L -25.4 °C, TBA-L treated -26.1°C; TBA-P -20.4 °C, TBA-P treated -20.9 °C; TBA-S -17.8 °C, TBA-S treated -18.2 °C) or K(-34 °C) values (TBA-L $3.5*10^7$ $mg^{-1}$, TBA-L treated $4.1*10^7$ $mg^{-1}$; TBA-P $2.2*10^8$ $mg^{-1}$, TBA-P treated $1.5*10^8$ $mg^{-1}$; TBA-S $2.4*10^8$ $mg^{-1}$, TBA-S treated $1.8*10^8$ $mg^{-1}$)."*

and with the corresponding Figure 5:

[Figure]

**Figure 1: Results of the heat treatment of the different TBA extracts. Leaves are marked with green circles, primary wood with violet stars and secondary wood with orange triangles. The left value belongs to the untreated sample, the right value to the sample treated with 100 °C for an hour. Filled symbols represent the mean freezing temperature and correlate with the left Y-axis, half filled symbols represent the cumulative nucleus concentration as -34 °C per mg extracted sample and correlate with the right Y-axis.**

and in the discussion section about the similarities to birch pollen washing water (see p8, l33-39).

*"The freezing temperature observed for the aqueous birch pollen extract (-17.1 °C see Figure 2), is in line with values reported in the literature for aqueous birch pollen extracts (reported freezing events are generally between -15 and -23 °C (Diehl et al., 2001; Pummer et al., 2012; Augustin et al., 2013; O'Sullivan et al., 2015)). Interestingly, most of our samples froze in that temperature range between -15 °C and -23 °C. Half of the leaves (TBC-L, TBD-L, TBF-L, TBG-L, and VB), eight out of ten primary wood samples (TBA-P, TBB-P, TBC-P, TBE-P, TBF-P, TBG-P, TBI-P, and TBV-P), and all secondary wood samples exhibited a mean freezing temperature in this temperature window. Moreover, we observed heat resistance at 100 °C, similar to the results of Pummer et al. (2012)."*

*b.) Another issue I would like to see addressed with regard to the nature of the INM is whether they could be a form of cellulose. This issue could be discussed with reference to the FTIR spectra in Figure 5 and also with regard to the slope of the cumulative nucleus spectra (Figure 3), as compared to similar spectra available for cellulose (e.g. Hiranuma et al., 2015, doi:10.1038/ngeo2374).*

Response: As the INP are contained in quite low concentrations, it is challenging to use the FTIR spectra for qualitative analytics of the INP. However, we added a section concerning the possible identity of the INP in the discussion section (p11, l14-24).

*"Only little INP are known to trigger freezing above -10°C, which are typically biological substances such as bacteria (Murray et al., 2012). Below -10 °C, birch pollen belong to the group of highest freezing temperatures, with onset higher than most mineral dusts, ash and soot samples (Murray et al., 2012). The vast majority of atmospheric INP and INP retrieved from precipitation samples exhibit freezing temperatures below -10°C (DeMott et al., 2010; Petters and Wright, 2015). The identity of the INP released from birches is still unclear. Pummer et al. (2013) showed that proteins, saccharides, and lipids are easily extracted aqueously from birch pollen. While Pummer et al. (2012) and Dreischmeier et al. (2017) speculate that the molecules are carbohydrates, Tong et al. (2015) attributes the highest INA to extracted proteins. Hiranuma et al. (2015) showed that cellulose, which is ubiquitous in plants, exhibits INA in the right temperature range. With our spectroscopic data, we found strong indicators for saccharides being present, including prominent bands which could be associated with cellulose. Further, we found bands in the most prominent protein regions, though those could be assigned to other molecule groups."*

*c.) In the Discussion you write that INM could be ": : :washed into the soil during rainfall: : :" (page 7, lines 29-30). Leaves and twigs are usually covered by a thin layer of wax to protect against desiccation. I wonder whether INM sitting in the tissues below the protective outer layer could be washed off. Wouldn't leaves and twigs first need to be shed and to disintegrate for INM to be washed off in larger numbers?*
Response: We changed this into *"Cracks and wounds on the surface could allow the INP to be washed of the surface of twigs and leaves into the soil. This marks a potential to influence the INA of mineral dust and soil particles and act as INP in the atmosphere."* (p8, l26-28) and added a small discussion on the importance of further studies on this topic. (see p8, l 30-31)

*"Further studies on possible release pathways of the INP from birches into the surrounding environment are necessary to quantify such effects."*

*d.) In Section 2.1 you introduce the altitudinal gradient along which you sampled the trees. Later in the paper there seems no further reference to this gradient. Instead, you relate results to the proximity of the trees to road or river. Is altitude irrelevant for the production of INM? Could similarity in terms of INM in a particular kind of location result from a genetic proximity of the trees (i.e. seeds spreading along a road or a river)?*
Response: We found no correlation between altitude and INM production. We expanded the discussion on this point based on your suggestions (see p9, l28):

*"Other than roads and rivers in close proximity, the tree altitude was not correlated to INA."*

*Page 2, line 9: Please be more precise. Concentrations reported by Christner et al. (2008) were quite low (at -10 C: 4 to 490 INP/L) compared to other studies (up to 500'000 INP/L at -10 C; Petters and Wright, 2015, dx.doi.org/10.1002/2015GL065733). What the paper by Christner et al. (2008) indeed has clearly shown was the large fraction (95%) of biological INP in the total number of INP.*
Response: This has been changed and the Petters and Wright citation has been included (p 2, l 6-11)

*"Precipitation can contain large amounts of INP. Petters and Wright (Petters and Wright, 2015) combined data from a large number of measurements and found a high variability in concentration in the range between -5 and -12 °C, which is assumed to be biological, with a maximum of approx. 500*

000 per L water. Christner et al. (2008) analysed snow and rain samples from the United States (Montana and Louisiana), the Alps and the Pyrenees, Antarctica (Ross Island) and Canada (Yukon), where they found rather low INP concentrations, but biological INP to represented the majority of the contained INP."

*Page 2, line 20: 'mechanism' seems more appropriate here than 'tool' (same in line 35).*
Response: The suggested changes have been implemented.

*The term "tissue" you use to denominate your samples does not seem correct to me. As I understand, you processed entire leaves and sections of twigs and branches, which you call primary and secondary wood. Branches, for example, are made up of several types of tissue (xylem, phloem, sclerenchyma, cortex, epidermis). I would find it more appropriate to not talk about "tissue" in your context but say that you analysed material from different parts of the trees (leave, twig, branch).*
Response: The suggested changes have been implemented and the terms have been changed throughout the manuscript.

*Trees differ in MFT and cumulative nucleus concentration in leaves. How reproducible are these values? Did you prepare and analyse, perhaps during the preparatory phase of your study, two or more samples from the same tree, i.e. did you process from one or several trees two sets of leaves or two sets of twig material?*
Response: We analysed a second branch from the birch TBA according to the described protocol. While we found minor differences for the primary and secondary wood samples, we found a significantly enhanced freezing temperature in the leaves of the second twig. The concentration of INP in the leaves remained constant. We included this in our results section (p6, l28-34):

*"To examine the INP distribution within a tree, a second branch of TBA was prepared and measured according to the described protocol. Resulting data are presented in Figure 2 and marked with a 2 (TBA-L2, TBA-P2, and TBA-S2). Primary and secondary wood extracts are well in line regarding their freezing temperatures (TBA P -20.4 °C, TBA-P2 -19.8 °C; TBA-S -17.8 °C, TBA-S2 -16.7 °C), however, the primary wood from the second analysed branch contained higher INP concentrations (TBA P 2.2\*10$^8$ mg$^{-1}$, TBA-P2 1.5\*10$^9$ mg$^{-1}$; TBA-S 2.4\*10$^8$ mg$^{-1}$, TBA-S2 3.7\*10$^8$ mg$^{-1}$). Leaves varied in their freezing temperatures and cumulative nucleus concentrations (TBA-L -25.3 °C and 3.5\*10$^7$ mg$^{-1}$, TBA-L2 -21.8 and 1.0\*10$^8$ mg$^{-1}$)."*

as well as in Figure 2:

[Figure]

**Figure 2: Top panel: Mean freezing temperature (MFT) of the different birch samples. Leaf extracts (L) are marked with a green circle, primary wood extracts (P) with a violet triangle, and secondary wood extracts (S) with an orange star. Further we introduced a dashed line for the MFT of ultrapure water (as a summary of regular measurements conducted over the course of the analysation of the presented samples, -36.2 °C, with a standard deviation of 0.5 °C (not plotted)), and a dotted line for the MFT of birch pollen washing water (-17.1°C with a standard deviation of 0.5 °C (not plotted)). The last three values on the right side represent the average of all mean freezing temperatures for leaves (AVG-L), primary wood (AVG-P) and secondary wood (AVG-S) with the corresponding standard deviation. Bottom panel: cumulative nucleus concentration at -34°C (K(-34 °C)) of the different birch samples per mg extracted sample. Assignment of the symbols is similar to the MFT plot. The dotted line refers to the K(-34 °C) of birch pollen washing water per mg extracted pollen ($1.3*10^{10}$ mg$^{-1}$). The last three values on the right side represent the average of all K(-34 °C) values. Error bars point to the area of trust, ranging from the highest to the lowest measured values.**

and the discussion section for the variability of the INA of leaves (p9, l5-6).

*"We observed a high variability of INM in leaves. Even for leaves of two branches of the same tree, we found differences in their freezing temperatures."*

**Further changes:**

We excluded the Saxena reference in the introduction

Figure 2 was split into 2 panels. Further we included the K(-34 °C) per mg birch pollen as reference line (introduced in p6, l20-22)

*"The dotted line in the lower panel refers to the K(-34 °C) value of birch pollen washing water (1.3\*10$^{10}$ mg$^{-1}$). Presented data shows that the samples with the highest K(-34 °C) values (TBB-S, and all samples from the Viennese birch) contain similar amounts of INP per mg extracted sample."*

We further included Sheil 2018 in the introduction (p 2, l 20-23)

*"While we know that forests influence the atmospheric water-cycle, the underlying processes are only poorly understood and characterized and it is important to further our understanding in this area, not just to enhance climatic predictions, but also to better understand the consequences of the changes in Earth's forests due to human activities (Sheil, 2018)."*

**Refrences:**

Augustin, S., Wex, H., Niedermeier, D., Pummer, B., Grothe, H., Hartmann, S., Tomsche, L., Clauss, T., Voigtländer, J., Ignatius, K. and Stratmann, F.: Immersion freezing of birch pollen washing water, Atmos. Chem. Phys., 13, 10989–11003, doi: 10.5194/acp-13-10989-2013, 2013.

Christner, B. C., Cai, R., Morris, C. E., McCarter, K. S., Foreman, C. M., Skidmore, M. L., Montross, S. N. and Sands, D. C.: Geographic, seasonal, and precipitation chemistry influence on the abundance and activity of biological ice nucleators in rain and snow, PNAS, 105(48), 18854–18859, doi: 10.1073/pnas.0809816105, 2008.

DeMott, P. J., Prenni, A. J., Liu, X., Kreidenweis, S. M., Petters, M. D., Twohy, C. H., Richardson, M. S., Eidhammer, T. and Rogers, D. C.: Predicting global atmospheric ice nuclei distributions and their impacts on climate, PNAS, 107(25), 11217–11222, doi: 10.1073/pnas.0910818107, 2010.

Diehl, K., Quick, C., Matthias-Maser, S., Mitra, S. K. and Jaenicke, R.: The ice nucleating ability of pollen Part I: Laboratory studies in deposition and condensation freezing modes, Atmos. Res., 58(2), 75–87, doi: 10.1016/S0169-8095(01)00091-6, 2001.

Dreischmeier, K., Budke, C., Wiehemeier, L., Kottke, T. and Koop, T.: Boreal pollen contain ice-nucleating as well as ice-binding "antifreeze" polysaccharides, Sci. Rep., 7(41890), doi: 10.1038/srep41890, 2017.

Hiranuma, N., Möhler, O., Yamashita, K., Tajiri, T., Saito, A., Kiselev, A., Hoffmann, N., Hoose, C., Jantsch, E., Koop, T. and Murakami, M.: Ice nucleation by cellulose and its potential contribution to ice formation in clouds, Nat. Geosci., 8(4), 273–277, doi: 10.1038/ngeo2374, 2015.

Murray, B. J., O'Sullivan, D., Atkinson, J. D. and Webb, M. E.: Ice nucleation by particles immersed in supercooled cloud droplets, Chem. Soc. Rev., 41(19), 6519–6554, doi: 10.1039/c2cs35200a, 2012.

O'Sullivan, D., Murray, B. J., Ross, J. F., Whale, T. F., Price, H. C., Atkinson, J. D., Umo, N. S. and Webb, M. E.: The relevance of nanoscale biological fragments for ice nucleation in clouds, Sci. Rep., 5(8082), 1–7, doi: 10.1038/srep08082, 2015.

Petters, M. D. and Wright, T. P.: Revisiting ice nucleation from precipitation samples, Geophys. Res. Lett., 42(20), 8758–8766, doi: 10.1002/2015GL065733, 2015.

Pummer, B. G., Bauer, H., Bernardi, J., Bleicher, S. and Grothe, H.: Suspendable macromolecules are responsible for ice nucleation activity of birch and conifer pollen, Atmos. Chem. Phys., 12, 2541–2550, doi: 10.5194/acp-12-2541-2012, 2012.

Pummer, B. G., Bauer, H., Bernardi, J., Chazallon, B., Facq, S., Lendl, B., Whitmore, K. and Grothe, H.: Chemistry and morphology of dried-up pollen suspension residues, J. Raman Spectrosc., 44(12), 1654–1658, doi: 10.1002/jrs.4395, 2013.

Sheil, D.: Forests , atmospheric water and an uncertain future : the new biology of the global water cycle, For. Ecosyst. 5, 19, doi: 10.1186/s40663-018-0138-y, 2018.

Tong, H. J., Ouyang, B., Nikolovski, N., Lienhard, D. M., Pope, F. D. and Kalberer, M.: A new electrodynamic balance (EDB) design for low-temperature studies: Application to immersion freezing

of pollen extract bioaerosols, Atmos. Meas. Tech., 8(3), 1183–1195, doi: 10.5194/amt-8-1183-2015, 2015.

---

## Author Comment (AC2) · 16 Apr 2018

The authors would like to thank the referee for the time and effort in reviewing our manuscript „Birch leaves and branches as a source of ice nucleating macromolecules"

We would like to discuss two general points brought up by the reviewer:

**1) The usage of fluorescence and IR spectroscopy as methods of characterization**

Fluorescence and IR spectroscopy have various upsides, including the fast measurements and very little requirements on samples and preparation and are therefore often applied, especially on complex systems as biological samples. Especially IR spectroscopy is widely used in literature since it gathers detailed molecular information related to the chemical reactivity and biological activity of the samples. This is especially true for literature on the ice nucleation activity of birches and birch pollen. We included a paragraph on the usage of IR spectroscopy and fluorescence spectroscopy in the introduction section (p3, l3-10)

*"Spectroscopic methods are a key instrument in characterizing complex biological systems. One of the methods typically applied on biological materials is infrared spectroscopy (Baker et al., 2015). Infrared spectroscopy can be applied for the characterization and discrimination of plants (Kim et al., 2004; Gorgulu et al., 2007; Anilkumar et al., 2012; Carballo-Meilan et al., 2014). Further infrared spectroscopy has already shown to respond well on the biochemical features of pollen of different species, allows differentiation of such (Gottardini et al., 2007; Pummer et al., 2013; Zimmermann and Kohler, 2014; Bağcioğu et al., 2015) and can even be used to gain information on the environmental conditions (Zimmermann and Kohler, 2014). While fluorescence spectroscopy is currently not used to discriminate different species, Pöhlker et al. (2013) showed that discrimination of pollen is possible with this technique on a family level."*

**2) The atmospheric impact of our work**

We included a paragraph about the possible atmospheric impact concerning plant debris and number concentrations of the INP of birch trees in the discussion (p9, l31-41).

"Some investigations on birch stands showed a dry weight of 2 to 25 t per ha for twigs and 1 to 8 t per ha for leaves (Johansson, 1999; Uri *et al.*, 2007). This leads to estimated INP concentrations on the order of $10^{16}$ to $10^{19}$ per ha for twigs and $10^{14}$ to $10^{18}$ per ha for leaves. Plant debris can be an important constituent of ambient particulate matter (Matthias-Maser and Jaenicke, 1995; Andreae, 2007; Winiwarter *et al.*, 2009). However, the underlying processes of the release of plant debris in the atmosphere is not fully understood, making predictions of their atmospheric impact hard (Andreae, 2007; Winiwarter *et al.*, 2009). Sánchez-Ochoa and colleagues analysed atmospheric aerosols collected at various background sites in Europe and used cellulose as a proxy for plant debris. They found biannual average concentrations of 33.4 to 363 ng per $m^3$ air (Sánchez-Ochoa *et al.*, 2007). Especially the leaves of birch trees could be an important source for INP as it is shed and produces annually. Decaying leaf litter is known to be a good source of INP (R.C. Schnell and Vali, 1973). Conen et al. (2016, 2017) showed that air masses passing over land can be enriched with INP derived from such leaf litter. Collectively, these studies underscore the importance of plants as sources of INP."

To further document our fit into this journal, we would like to point to the sizeable number of papers published in ACP concerning primary biological aerosols and their impact on heterogeneous ice nucleation (as e.g. Huffman et al. 2013 10.5194/acp-13-6151-2013, Hummel et al. 2018 10.5194/acp-2018-182), as well as submicron biological INP (Pummer et al. 2012 10.5194/acp-12-2541-2012, Augustin et al. 2013 10.5194/acp-13-10989-2013, Pummer et al. 2015 10.5194/acp-15-4077-2015), and the influence of biological residues on the ice nucleation activity of other particles (Conen et al. 2011 10.5194/acp-11-9643-2011, Tobo et al. 2014 10.5194/acp-14-8521-2014, Hill et al. 2016 10.5194/acp-16-7195-2016, O'Sullivan et al. 2016 10.5194/acp-16-7879-2016). We think that our data can contribute to all of these fields. Also we would like to stress the point that our data indicate that there could be a greater fraction of heat resistant biological INP in the atmosphere suggesting that heat treatment alone is not sufficient to completely discriminate between biological and non-biologial INP.

*Pg 3, line20 –The authors removed "visible" contamination such as lichen. How might leaving "sub-visible" contamination affect the outcomes? I would think that removing only the obvious layer could include still significant amounts of nuclei that could still influence results. Alternatively, by taking the same sample and stripping the outer bark so that there was no possible contamination between external molecules (whether lichen, deposited pollutants, etc.) could isolate this issue.*

Response: As the samples used are of natural origin, we have to assume impurities to be present. However, since we do not know the distribution of the INP throughout the tissue and the role of the bark in this process, stripping samples could affect the outcome tremendously without pointing to the role of impurities per se. Especially problematic are the secondary wood samples, which often exhibit a rough fractured surface and we would need to strip not just the bark, but also the outermost layers of wood contained underneath to ensure the removal of all layers, which were in contact with the surrounding environment.

The centrifugation and filtration helps minimize the possible effect impurities can have on our samples, as most biological and mineral material, which is known to be ice nucleation active will not pass through the 0.2 μm syringe filter (we added a remark about this in the sample preparation section (p4, l4-7) *"Afterwards it was centrifuged (3500 rpm/ 1123 g for 5 min) and the supernatant was pressed through a 0.2 μm syringe filter (VWR, cellulose acetate membrane, sterile), removing all bigger particles, as well as possible impurities from e.g. intact bacterial cells."*). Especially biological material is in some cases known to release INP into the aqueous phase, which are in the submicron size range. These INP however, were shown to trigger freezing at temperatures typically above -10°C. As we did not observe a single freezing event at such high temperatures, we assume biological impurities to be of minor importance in our samples. To further address this important problem, we added another paragraph to the discussion (p10, l1-13) to discuss the possible role of impurities on our samples.

*"Since all of the analysed materials are of natural origin, we cannot rule out that some contamination could play a role in the INA of our extracts. Some bacteria have been found to act as INP (as e.g. Pseudomonas syringae (Maki et al., 1974)), however, these bacteria are typically in the size range > 1 μm (Monier and Lindow, 2003) and therefore easily filtered with the 0.2 μm syringe filter. Further, some lichen are known to be INA (Kieft, 1988), and some microorganisms release their small contained INP in the aqueous phase as e.g. Mortierella alpine (Fröhlich-Nowoisky et al., 2015), which cannot be filtered with used methods. However, most known ice nucleation active lichens and microorganisms as well as released INP typically freeze at significantly higher temperatures (above -10°C (Maki et al., 1974; Kieft, 1988; Pouleur et al., 1992; Murray et al., 2012; Fröhlich-Nowoisky et al.,*

*2015) than the freezing temperatures observed for our samples, with very little exceptions (Iannone et al., 2011). As the highest onset temperature observed in our measurements was -14.1 °C (TBC-L), and the onset temperature of birch pollen washing water was quite close to this value (-15.1 °C), and heat treatment did not affect the extracts of TBA, we do not suspect significant contamination of our samples. However, the INA of birches, especially if growing close to a road or in urban regions, could be affected by soot and other anthropogenic emissions, as soot can act as INP (DeMott, 1990; Murray et al., 2012)"*

*Pg 3, line27–The drying process was continued until the weight was constant. How did the authors define "constant?"*

Response: As weight consistency between two weight measurements separated by at least 2 h with ongoing drying procedure between the measurements. This has been included in the methodology part (p3-4, l41-1).

*"All samples were dried for at least twelve hours. Weight consistency was determined by two weighing steps separated by at least two hours of drying."*

*Pg5, related to Fig. 2 – Since the authors draw conclusions about the types of birch material (leaves, primary wood, etc.), it would be good to show averages + std dev of each type on either the left or right within this figure.*

Response: The suggested changes have been implemented in Figure 2.

*Pg 6, line33 – "pointing to the importance of polysaccharides in our extracts" This is an example of an overstatement, in my opinion. While the polysaccharides may include these specific infrared bands, fundamentally these are vibrational features of individual chemical bonds that can exist in many types of molecules.*

Response: The statement has been removed. We included a short discussion on polysaccharides (p10, l 15-18)

*"The measured FTIR spectra indicate that the birch extracts are chemically similar to each other, and to pure birch wood. As plants do not only contain polysaccharides but several soluble carbohydrates (Magel et al., 2000), we assume those substances to play an important role in the chemical composition of our extracts. Fitting to this assumption, most of the bands found in our spectra could be assigned to carbohydrates and polysaccharides."*

*Pg 6, line 34 – "can be assigned to other biomolecules" . . . similar to the comment above. I think it would be better stated as "are consistent with" in place of "can be assigned to"*

Response: This has been changed.

*Pg 7, Section 3.4 – Subtle differences in intensity of fluorescence peaks here could easily be a function of analyte concentration. How did the authors control for concentration? If the authors are suggesting that the 10% differences in the peak heights (e.g. of the 260 nm Ex) are due to chemical or biological differences in the sample, they should discuss how they are confident it is not just subtle dilution effects.*

Response: Unfortunately, we cannot control the analyte concentration, as the contained mixture in the different extracts is too diverse to be easily assessed. However, we do not suggest that this is due to chemical or biological differences. Quite on the contrary, we believe our spectra show quite well that none of the analysed extracts contain fluorescent analytes active in the observed range, which

cannot be found in all other extracts too. We broadened the discussion on this point (see p10, l34-37)

*"Throughout the different extracts we found the same peaks, which might stem from similarities in fluorescent analytes between pollen and branch extracts. Small differences in intensities and ratios could result from differences in the concentration of the active substances"*

*Pg7, line 24 – "Most of our samples froze at temperatures close to the freezing temperature of birch pollen washing water." This line is a bit vague. What do the authors mean by "close to" here and "most?"*
Response: We specified this paragraph (see p8, l33-39).

*"The freezing temperature observed for the aqueous birch pollen extract (-17.1 °C see Figure 2), is in line with values reported in the literature for aqueous birch pollen extracts (reported freezing events are generally between -15 and -23 °C (Diehl et al., 2001; Pummer et al., 2012; Augustin et al., 2013; O'Sullivan et al., 2015)). Interestingly, most of our samples froze in that temperature range between -15 °C and -23 °C.). Half of the leaves (TBC-L, TBD-L, TBF-L, TBG-L, and VB), eight out of ten primary wood samples (TBA-P, TBB-P, TBC-P, TBE-P, TBF-P, TBG-P, TBI-P, and TBV-P) and all secondary wood samples exhibited a mean freezing temperature in this temperature window. Moreover, we observed heat resistance at 100 °C, similar to the results of Pummer et al. (2012)."*

*Pg 8, line33 – "show strong similarities .. shown by Chen et al." Can the authors expand the discussion on this point? After looking up the spectra shown by Chen et al., I was a bit confused. I see that the Chen spectra seem to be somewhat higher in resolution, but otherwise I wasn't sure what specific points the authors were trying to extract from the comparison.*
Response: The main reason to include this citation was the comparison between a pure wood sample and our aqueous extracts. With this we like to show that our extracts exhibit most IR spectroscopic patterns found in pure wood by other working groups except for lignin, which is only very weakly soluble in water. Therefore these differences were expected. We tried to make the point of this comparison clearer (see p10, l25-28)

*"Other than the lignin bands, our aqueous extracts show very similar spectroscopic features compared to the pure wood samples. These similarities between the spectra of our extracts and the spectrum of pure wood indicate that our extract method retrieves the majority of components, leading to a similar distribution of bands, with differing intensities due to differences in concentration."*

*Pg 8, first paragraph – How would these spectra look if you did the same with material from other tree species? Fluorescence spectra are always broad (i.e. compared to IR spectra), and then when grinding large volumes of material to be mixed into a sample for a spectrum – the analysis is obviously very homogeneous and mixed with huge numbers of types of molecules. It does not surprise me that these four sets of spectra look similar – it would surprise me if they looked very different. In contrast, I would expect the same spectra from another tree species to look very similar, so it is hard to know what this fluorescence spectra adds to the overall analysis in the manuscript. Can the authors provide comparisons to fluorescence spectra published elsewhere? Surely this has been done and is otherwise reported.*
Response: Pollen of several tree species have been analysed by Pöhlker et al. 2013 (10.5194/amt-6-3369-2013) and showed that the different species can be differentiated on a family level by maxima and relation between maxima. In our presented fluorescence data, the maxima of the different tree extracts and the birch pollen extracts look very similar not just in the position of the maxima but also

in the relation between the different maxima. The only exception is the primary wood showing a slightly enhanced peak maximum at the 260 nm excitation wavelenght.

*Page 9, line 24 – "suggest that birch tissues tested contained chemical substances similar to birch pollen." I disagree with the weight of this statement. I think that the results suggest that the samples may have exhibited broadly similar IR and fluorescence spectral features, but to extend the statement to say that the "chemical substances" were similar was never tested directly here. Also, the data shown in the paper suggest that spectra from different types of material from the same plant are relatively similar, but differences across plant samples are not directly shown.*
Response: We changed this to *"aqueous extracts of birch materials tested showed similarities to aqueous extracts of birch pollen"* (p11, l35-36) Further we included another paragraph in the discussion section about similarities between the different samples (p11, l7-12)
*"In both, FTIR and fluorescence spectroscopy, we found strong similarities between birch pollen washing water and the different aqueous extracts from the TBA samples. Further comparison with whole pollen grains (for both FTIR and fluorescence spectroscopy), as well as with pure wood (for FTIR), as found in literature, shows strong similarities in the spectroscopic features of our different birch samples. As not just the band position, but also the intensity ratios are agreeable with each other, we assume this to indicate that we are able to extract the major components found in wood with our extraction method and that the pollen and wood samples extracts exhibit chemical similarities to a certain extend."*

*Figures – In general, I would suggest using color for figures 2-4. For Fig 2, I would also put the circle/triangle/star detail into the figure legend, and not just in the caption. This would make the complex figure easier to read.*
Response: The suggested changes have been implemented.

[Figure]

Figure 1: Top panel: Mean freezing temperature (MFT) of the different birch samples. Leaf extracts (L) are marked with a green circle, primary wood extracts (P) with a violet triangle, and secondary wood extracts (S) with an orange star. Further we introduced a dashed line for the MFT of ultrapure water (as a summary of regular measurements conducted over the course of the analysation of the presented samples, -36.2 °C, with a standard deviation of 0.5 °C (not plotted)), and a dotted line for the MFT of birch pollen washing water (-17.1°C with a standard deviation of 0.5 °C (not plotted)). The last three values on the right side represent the average of all mean freezing temperatures for leaves (AVG-L), primary wood (AVG-P) and secondary wood (AVG-S) with the corresponding standard deviation. Bottom panel: cumulative nucleus concentration at -34°C (K(-34 °C)) of the different birch samples per mg extracted sample. Assignment of the symbols is similar to the MFT plot. The dotted line refers to the K(-34 °C) of birch pollen washing water per mg extracted pollen (1.3*10^10 mg^-1). The last three values on the right side represent the average of all K(-34 °C) values. Error bars point to the area of trust, ranging from the highest to the lowest measured values.

[Figure]

**Figure 2: Cumulative nucleus concentration as a function of temperature for leaf extracts (right), primary wood extracts (middle), and secondary wood extracts (right). The diagram is cut off at -35°C, since we cannot contribute freezing events below this temperature to heterogeneous nucleation. The symbols used for the different data points are grouped. Birches growing in close proximity under similar conditions are marked with the same symbol (different fillings).**

[Figure]

**Figure 3: Scatterplot of dry mass (dry residues of the different filtered extracts) and cumulative nucleus concentration at -34°C per sample mass. The dry mass is the mass we were able to extract with the 50 mg/mL suspensions. The data show that secondary wood, which contained mostly the highest INM concentrations and lowest variations between different samples, also contained the lowest extractable mass. Therefore INM ratios in the extractable content of the different samples were highest in secondary wood samples.**

*Figure 3 – How do these data compare to other atmospheric measurements using*

Response: We compared our data to the freezing temperature range of other known atmospheric INP and precipitation samples in the discussion section (p11, l14-24), where we compare our results to different substances and try to sum up what is known about the identity of the birch pollen derived INP.

*"Only little INP are known to trigger freezing above -10°C, which are typically biological substances such as bacteria (Murray et al., 2012). Below -10 °C, birch pollen belong to the group of highest freezing temperatures, with onset higher than most mineral dusts, ash and soot samples (Murray et al., 2012). The vast majority of atmospheric INP and INP retrieved from precipitation samples exhibit freezing temperatures below -10°C (DeMott et al., 2010; Petters and Wright, 2015). The identity of the INP released from birches is still unclear. Pummer et al. (2013) showed that proteins, saccharides, and lipids are easily extracted aqueously from birch pollen. While Pummer et al. (2012) and Dreischmeier et al. (2017) speculate that the responsible molecules are carbohydrates, Tong et al. (2015) attributes the highest INA to extracted proteins. Hiranuma et al. (2015) showed that cellulose, which is ubiquitous in plants, exhibits INA in the right temperature range With our spectroscopic data, we found strong indicators for saccharides being present, including prominent bands which could be associated with cellulose. Further, we found bands in the most prominent protein regions, though those could be assigned to other molecule groups."*

**Further changes:**

We excluded the Saxena reference in the introduction

Figure 2 was split into 2 panels. Further we included the K(-34 °C) per mg birch pollen as reference line (introduced in p6, l20-22)

*"The dotted line in the lower panel refers to the K(-34 °C) value of birch pollen washing water ($1.3*10^{10}$ $mg^{-1}$). Presented data shows that the samples with the highest K(-34 °C) values (TBB-S, and all samples from the Viennese birch) contain similar amounts of INP per mg extracted sample."*

We further included Sheil 2018 in the introduction (p 2, l 20-23)

*"While we know that forests influence the atmospheric water-cycle, the underlying processes are only poorly understood and characterized and it is important to further our understanding in this area, not just to enhance climatic predictions, but also to better understand the consequences of the changes in Earth's forests due to human activities (Sheil, 2018)."*

**Refrences:**

Andreae, M. O.: Aerosols before pollution, Science, 315(5808), 50–51, doi: 10.1126/science.1136529, 2007.

Anilkumar, V. S., Dinesh Babu, K. V, Sunukumar, S. S. and Murugan, K.: Taxonomic discrimination of Solanum nigrum and S. giganteum by Fourier transform infrared spectroscopy Data, J. Res. Biol., 2(5), 482–488, , 2012.

Augustin, S., Wex, H., Niedermeier, D., Pummer, B., Grothe, H., Hartmann, S., Tomsche, L., Clauss, T., Voigtländer, J., Ignatius, K. and Stratmann, F.: Immersion freezing of birch pollen washing water, Atmos. Chem. Phys., 13, 10989–11003, doi: 10.5194/acp-13-10989-2013, 2013.

Bağcioğlu, M., Zimmermann, B. and Kohler, A.: A multiscale vibrational spectroscopic approach for identification and biochemical characterization of pollen, PLoS One, 10(9), 1–19, doi: 10.1371/journal.pone.0137899, 2015.

Baker, M. J., Trevisan, J., Bassan, P., Bhargava, R. and Butler, H. J.: Using Fourier transform IR spectroscopy to analyze biological materials, Nat Protoc, 9(8), 1771–1791, doi: 10.1038/nprot.2014.110.Using, 2015.

Carballo-Meilan, A., Goodman, A. M., Baron, M. G., and Gonzalez-Rodriguez, J.: A specific case in the classification of woods by FTIR and chemometric: Discrimination of Fagales from Malpighiales, Cellulose, 21(1), 261–273, doi: 10.1007/s10570-013-0093-2, 2014.

Conen, F., Stopelli, E., and Zimmermann, L.: Clues that decaying leaves enrich Arctic air with ice nucleating particles, Atmos. Environ., 129, 91–94, doi: 10.1016/j.atmosenv.2016.01.027, 2016.

Conen, F., Yakutin, M. V, Yttri, K. E. and Hüglin, C.: Ice Nucleating Particle Concentrations Increase When Leaves Fall in Autumn, 8(202), 1–9, doi: 10.3390/atmos8100202, 2017.

DeMott, P. J., Prenni, A. J., Liu, X., Kreidenweis, S. M., Petters, M. D., Twohy, C. H., Richardson, M. S., Eidhammer, T. and Rogers, D. C.: Predicting global atmospheric ice nuclei distributions and their impacts on climate, PNAS, 107(25), 11217–11222, doi: 10.1073/pnas.0910818107, 2010.

DeMott, P. J.: An Exploratory Study of Ice Nucleation by Soot Aerosols, J. Appl. Meteorol., 29, 1072–1079, doi: 10.1175/1520-0450(1990)029<1072:AESOIN>2.0.CO;2, 1990.

Diehl, K., Quick, C., Matthias-Maser, S., Mitra, S. K. and Jaenicke, R.: The ice nucleating ability of pollen Part I: Laboratory studies in deposition and condensation freezing modes, Atmos. Res., 58(2), 75–87, doi: 10.1016/S0169-8095(01)00091-6, 2001.

Dreischmeier, K., Budke, C., Wiehemeier, L., Kottke, T. and Koop, T.: Boreal pollen contain ice-nucleating as well as ice-binding "antifreeze" polysaccharides, Sci. Rep., 7(41890), doi: 10.1038/srep41890, 2017.

Fröhlich-Nowoisky, J., Hill, T. C. J., Pummer, B. G., Yordanova, P., Franc, G. D. and Pöschl, U.: Ice nucleation activity in the widespread soil fungus Mortierella alpina, Biogeosciences, 12, 1057–1071, doi: 10.5194/bg-12-1057-2015, 2015.

Gorgulu, S. T., Dogan, M. and Severcan, F.: The characterization and differentiation of higher plants

by Fourier transform infrared spectroscopy, Appl. Spectrosc., 61(3), 300–308, doi: 10.1366/000370207780220903, 2007.

Gottardini, E., Rossi, S., Cristofolini, F. and Benedetti, L.: Use of Fourier transform infrared (FT-IR) spectroscopy as a tool for pollen identification, Aerobiologia (Bologna)., 23(3), 211–219, doi: 10.1007/s10453-007-9065-z, 2007.

Hiranuma, N., Möhler, O., Yamashita, K., Tajiri, T., Saito, A., Kiselev, A., Hoffmann, N., Hoose, C., Jantsch, E., Koop, T. and Murakami, M.: Ice nucleation by cellulose and its potential contribution to ice formation in clouds, Nat. Geosci., 8(4), 273–277, doi: 10.1038/ngeo2374, 2015.

Iannone, R., Chernoff, D. I., Pringle, A., Martin, S. T. and Bertram, A. K.: The ice nucleation ability of one of the most abundant types of fungal spores found in the atmosphere, Atmos. Chem. Phys., 11(3), 1191–1201, doi: 10.5194/acp-11-1191-2011, 2011.

Johansson, T.: Biomass equations for determining fractions of common and grey alders growing on abandoned farmland and some practical implications, Biomass and Bioenergy, 16, 223–238, doi: https://doi.org/10.1016/S0961-9534(98)00075-0, 1999.

[revised manuscript text omitted]

---

## Author Response (AR2)

The authors would like to thank referee #1 for the time and effort in reviewing our manuscript „Birch leaves and branches as a source of ice nucleating macromolecules"

The manuscript by Felgitsch et al. presents evidence regarding the ice-nucleating macromolecules (INM) in samples taken from several portions of birch trees in Austria. The revised manuscript improves clarity in a number of areas. My present concern is that the link between the evidence about ice nucleating temperature and particle number is almost totally decoupled from the spectroscopic evidence used to investigate the chemical nature of the samples. I think the data presented about various samples from the trees are interesting, and the authors have improved the manuscript by including at least hypothetical scenarios by which macromolecules within the tree wood could possibly contribute to soil or atmospheric ice nucleation.

Response: We do not claim a direct correlation between the spectra and the different INP concentrations, but a close proximity between the chemical components of birch pollen wash and the washes of the different analysed parts of the trees. It is important to note that we find accordance (in terms of band positions) between the different samples (leaves and wood). Therefore, the spectra are useful for sample comparisons, as opposed to detailed analyses of INM.

Line 17 of page 1 (abstract) and line 32 of page 11 (conclusions) state similarly that "the majority of the samples showing freezing temperatures close to those of birch pollen extracts, indicating a relationship between the INM of wood, leaves and pollen." I have two concerns with this statement. First, the meaning of "close to" needs to be clearly defined in both of these instances. At least four of the leaf samples and several primary wood samples are closer to ultrapure water than to the pollen wash water, so the text should clarify how the reader should interpret "close."

Response: Here we define what "close to" means for our analysis: A nonparametric analysis shows that the distributions of droplet freezing temperatures are similar among more than a quarter of all samples and pollen washing water as well as two dilutions thereof. The temperature regime of freezing events of birch pollen washing water broadens when it is diluted. This analysis is especially important as our samples all exhibit smaller INM concentration than pure washing water. When comparing our samples we were able to match some of the samples with the highest activity to pure washing water and samples with the lowest activity to a strong dilution thereof (1:10,000). We changed the two mentioned statements to:

"Concentrations of ice nuclei ranged from $6.7*10^4$ to $6.1*10^9$ per mg sample. Mean freezing temperatures varied between -15.6 °C and -31.3 °C; the range of temperatures where washes of birch pollen and dilutions thereof typically freeze. The freezing behaviour of three concentrations of birch pollen washing water (initial wash, 1:100, and 1:10,000), were significantly associated with more than a quarter of our samples, including some of the samples with highest and lowest activity, indicating a relationship between the INM of wood, leaves and pollen." (p1l15-20)

"Comparing the freezing behaviour of our samples to birch pollen washing water and two dilutions (1:100 and 1:10,000) using the Wilcoxon–Mann–Whitney test ,we found statistical correlations for more than a quarter of our samples and birch pollen washing water, indicating a relationship between the INM of wood, leaves and pollen. As we were able to match some of the most and least active samples, we conclude that given the right dilution of birch pollen washing water, all samples could be matched." (p11l12-15)

A more detailed explanation for these results is given in the answers below.

Second, without any presented statistics, it is hard to know how well to trust the second part of the statement, which uses the freezing temperature (i.e. Fig. 2/top) as support that INM from wood, leaves, and pollen are similar. This may or may not be true, but the evidence from Figure 2 does not

seem to strongly and systematically support that statement. The authors also discuss this same point at the bottom of page 8 (line 41) where after discussing Figure 2 they state that "Based on these results, we hypothesize that the INM in birch trees are quite similar in pollen, leaves, primary wood, and secondary wood.":.

Response: The INA of birch pollen washing water is highly depending on its concentration. If the concentration is lowered, we observe a broadening of the freezing events (between -15 °C and -35 °C) and a shift of the MFT to lower temperatures. To clarify this we introduced two dilutions of birch pollen washing water (1:100 simulates a sample with $10^8$ INP per mg and 1:10,000 simulates a sample with $10^6$ INP per mg). We altered Figure 2 accordingly and introduced new Figure 3:

[revised manuscript text omitted]

In contrast to their conclusions from the figure, I see a trend where the leaves are generally lower in freezing temperature than the other tree samples, the primary wood is frequently next coldest, and the secondary wood is slightly warmer still. This interpretation of the figure could be used to support a contrasting conclusion that INM from different fragments are *different*. Based on this important possible discrepancy I think the evidence is not presented to support the "indicating a relationship …" portion of the statement. If the statistical meaning of the terms are presented and supported, it could then be more defensible:

Concerning the differences in freezing temperature when comparing primary wood, secondary wood, and leaves, we introduced the following passage in the results section:

"The averages of MFT and cumulative nucleus concentration (Figure 2) show a similar trend. Leaves exhibit lowest freezing temperature and cumulative concentration, followed by primary wood, and secondary wood exhibit highest values in both categories. This points towards a relationship between concentration and freezing temperature as it has already been observed for the birch pollen extracts." (p6l30-34)

Further we altered the discussion section:

"The data shows that the average freezing temperatures of secondary wood, primary wood, and leaves differ slightly. These differences however follow the same pattern as the INM concentration. Therefore we assume this to be a concentration effect. Based on these results, we hypothesize that the INM in birch, which are found in trees are quite similar in pollen, leaves, primary wood, and secondary wood behave similar and can be statistically related to the INM found in birch pollen."(p8-9l39-1)

The authors also show IR and fluorescence spectra from fragments of tree TBA, but no data is shown for any other trees. If the authors wish to use the spectroscopic data to support any general conclusions, I do not see how this can be possible without systematically showing the data for all trees and for all sample types. Then the authors can attempt to generalize the results in some way. For example on page 10 (line 15) the authors state "The measured FTIR spectra indicate that the birch extracts are chemically similar to each other, and to pure birch wood." Again the definition of "similar" is important here, in part because the majority of the molecular composition may be similar, but it is hard to know how this may or may not relate to the INM. One would guess that the INM are likely a small minority of the overall molecular composition, and so I see very little link between the ice nucleating data and the spectroscopic data. Both are interesting independently, but trying to tie them together is more problematic in my opinion.
I think the manuscript is an interesting contribution to the overall scientific literature, but I suggest the scope of the discussion and conclusion statements be given more quantitative support before publication.

Response: We agree with the referee that the inclusion of spectra of all analysed samples is desirable. We put all infrared spectra in the supporting information and referred to this in p5l21:

"IR spectra of all other extracts can be found in the supporting information (see Figure S1-3)."

And in p8l2-3:

"The spectra of all analysed samples are given in the supplementary Figures S1-3. They show the same features as the spectra given for TBA, with varying intensity ratios." All three figures are given below.

Unfortunately, we did not have enough material to record FTIR and fluorescence spectra for all of the samples. Since FTIR spectra give more detailed information concerning the sample composition, we report the FTIR spectra in this publication. Fluorescence spectra are less informative, and the results are often clouded by overlapping of band positions. As already indicated in the manuscript and also proposed by us and by other authors, polysaccharides represent a major component in our samples (Pummer 2013, Dreischmeier 2017, Magel et al. 2000). FTIR is superior to fluorescence spectroscopy concerning the analysis of polysaccharides. In order to avoid showing a fragmentary set of data, we decided to withdraw the fluorescence data completely.

DePuy, V.; Berger, V.W.; Zhou, Y.: Wilcoxon–Mann–Whitney Test, Everitt, P.S., Howell, D.C. (Eds) Encyclopedia of Statistics in Behavioral Science, Vol. 4, 2118–2121, 2005.

Dreischmeier, K., Budke, C., Wiehemeier, L., Kottke, T. and Koop, T.: Boreal pollen contain ice-nucleating as well as ice-binding "antifreeze" polysaccharides, Sci. Rep., 7(41890), doi: 10.1038/srep41890, 2017.

Magel, E., Einig, W. and Hampp, R.: Carbohydrates in trees, Dev. Crop Sci., 26(C), 317–336, doi: 10.1016/S0378-519X(00)80016-1, 2000.

Pummer, B. G., Bauer, H., Bernardi, J., Chazallon, B., Facq, S., Lendl, B., Whitmore, K. and Grothe, H.: Chemistry and morphology of dried-up pollen suspension residues, J. Raman Spectrosc., 44(12), 1654–1658, doi: 10.1002/jrs.4395, 2013.

[Figure]

**Figure S1: The IR spectra of the all analysed extracts of leaves. Left: the whole spectral range, right: a close-up on the spectra below 1800 cm⁻¹.**

[Figure]

**Figure S2: The IR spectra of the all analysed extracts of primary woods. Left: the whole spectral range, right: a close-up on the spectra below 1800 cm⁻¹.**

[Figure]

**Figure S3: The IR spectra of the all analysed extracts of secondary woods. Left: the whole spectral range, right: a close-up on the spectra below 1800 cm⁻¹.**

[revised manuscript text omitted]

---

## Author Response (AR3)

The authors would like to thank the referee#2 for the time and effort in reviewing our manuscript „Birch leaves and branches as a source of ice nucleating macromolecules". Further we would like to thank the editor Dr. Ryan Sullivan for the quick processing of our final manuscript as well as for his quick responses in addressing all our questions and requests.

[revised manuscript text omitted]